



# Maize root and shoot litter quality controls short-term CO₂ and N₂O emissions and bacterial community structure of arable soil

Pauline Sophie Rummel[1], Birgit Pfeiffer[1,2], Johanna Pausch[3], Reinhard Well[4], Dominik Schneider[2], Klaus Dittert[1]

[1]Division of Plant Nutrition and Crop Physiology, Department of Crop Science, University of Göttingen, Germany
[2]Institute of Microbiology and Genetics, Dept. of Genomic and Applied Microbiology, University of Göttingen, Germany
[3]Agroecology, Faculty for Biology, Chemistry, and Earth Sciences, University of Bayreuth, Germany
[4]Thünen Institute, Climate-Smart Agriculture, Braunschweig, Germany

*Correspondence to*: Pauline Sophie Rummel (pauline.rummel@uni-goettingen.de)

**Abstract.** Chemical composition of root and shoot litter controls decomposition and, subsequently, C availability for biological nitrogen transformation processes in soils. While aboveground plant residues have been proven to increase N₂O emissions, studies on root litter effects are scarce. This study aimed 1) to evaluate how fresh maize root litter affects N₂O emissions compared to fresh maize shoot litter, 2) to assess whether N₂O emissions are related to the interaction of C and N mineralization from soil and litter, and 3) to analyze changes in soil microbial community structures related to litter input and N₂O emissions.

To obtain root and shoot litter, Maize plants (*Zea mays* L.) were cultivated with two N fertilizer levels in a greenhouse and harvested. A two-factorial 22-day laboratory incubation experiment was set up with soil from both N levels (N1, N2) and three litter addition treatments (Control, Root, Root+Shoot). We measured hourly CO₂ and N₂O fluxes, analyzed soil nitrate and water extractable organic C (WEOC) concentrations, and determined quality parameters of maize litter. Bacterial community structures were analyzed using 16S rRNA gene sequencing.

Maize litter quality controlled NO₃⁻ and WEOC availability and decomposition related CO₂ emissions. High bioavailability of maize shoot litter strongly increased CO₂ and N₂O emissions, while emissions induced by maize root litter remained low. We identified a strong positive correlation between cumulative CO₂ and N₂O emissions, supporting our hypothesis that litter quality affects denitrification by creating plant litter associated anaerobic microsites. The interdependency of C and N availability was validated by analyses of regression. Moreover, there was a strong positive interaction between soil NO₃⁻ and WEOC concentration resulting in much higher N₂O emissions, when both NO₃⁻ and WEOC were available. A significant correlation was observed between total CO₂ and N₂O emissions, the soil bacterial community composition and the litter level, showing a clear separation of Root+Shoot samples of all remaining samples. Bacterial diversity decreased with higher N level and higher input of easily available C. Altogether, changes in bacterial community structure reflected degradability of maize litter with easily degradable C from maize shoot litter favoring fast growing C cycling and N reducing bacteria of the phyla *Actinobacteria*, *Chloroflexi*, *Firmicutes* and *Proteobacteria*.



## 1 Introduction

Chemical composition controls decomposition of both roots (Birouste et al., 2012; Redin et al., 2014; Silver and Miya, 2001) and plant litter (Jensen et al., 2005; Kögel-Knabner, 2002; Zhang et al., 2008) and, subsequently, C availability for biological

nitrogen transformation processes in soils. When $O_2$ concentrations are low, denitrifying soil microorganisms may use nitrate ($NO_3^-$) as electron acceptor in the respiratory chain to break down organic compounds (Zumft, 1997). This leads to loss of plant available N (Müller and Clough, 2014) and makes soils an important source of the greenhouse gas $N_2O$ (Ciais et al., 2013).

Plant residues have been proven to increase $N_2O$ emissions upon incorporation into soil. When different types of litter were

compared, quality parameters of plant residues, such as C:N ratio, lignin:N ratio and chemical composition of structural components explained a large share of variances in $N_2O$ emissions (Baggs et al., 2000; Chen et al., 2013; Millar and Baggs, 2004). Especially in drier soils, denitrification is largely controlled by the supply of readily decomposable organic matter (Azam et al., 2002; Burford and Bremner, 1975; Loecke and Robertson, 2009). Availability of easily degradable C compounds stimulates microbial respiration, limiting $O_2$ at the microsite level and increasing $N_2O$ emissions (Azam et al.,

2002; Chen et al., 2013; Miller et al., 2008). Furthermore, plant litter enhances local anaerobicity by absorbing water from surrounding pores and retaining high moisture concentrations (Kravchenko et al., 2017, 2018).

While effects of aboveground plant residues on $N_2O$ emissions have been studied extensively, studies of root residues on $N_2O$ emissions are scarce. In a temperate forest soil, fine root litter of maize and native tree species did not cause any $N_2O$ emissions, but a very close interrelation between C mineralization of fine root litter and $N_2O$ emission was found in other

biomes (Hu et al., 2016). In other studies, lower cumulative $N_2O$ emissions were reported after addition of sugar beet roots compared to leaves (Velthof et al., 2002) and rice roots compared to rice straw (Lou et al., 2007). Furthermore, decomposition dynamics of roots have been studied in great detail, revealing that chemical composition explains most of its variation (Birouste et al., 2012; Johnson et al., 2007; Machinet et al., 2011; Redin et al., 2014; Silver and Miya, 2001; Zhang and Wang, 2015). In general, decomposition rates of hemicelluloses and pectin are higher than that of cellulose, while

among cell wall components lignin is most resistant against microbial decomposition (Kögel-Knabner, 2002).

Soil microorganisms are often specialized in specific substrates with fungi being regarded as the main decomposers of plant materials rich in cellulose and lignin, while hemicelluloses and pectin are decomposed by many aerobic and anaerobic bacteria and fungi (Kögel-Knabner, 2002). While the phyla *Firmicutes*, *Proteobacteria*, and *Bacteriodetes* are described as fast growing copiotrophic bacteria that are stimulated by input of easily degradable C compounds (Fierer et al., 2016;

Pascault et al., 2013), abundance of *Acidobacteria* decreased following the addition of dissolved organic matter into the soil (Fierer et al., 2016). Similarly, denitrifying microorganisms are found in bacteria, fungi and archaea depending on substrate availability and environmental conditions (Zumft, 1997). Fungi are seen as major contributors to denitrification under aerobic and weakly anaerobic conditions, while bacterial denitrification predominates under strongly anaerobic conditions (Hayatsu et al., 2008). Denitrifying bacteria can be found in most phyla (Zumft, 1997), with dominant populations in



*Pseudomonas* and *Alcaligenes* (Gamble et al., 1977; Megonigal et al., 2013). The most abundant denitrifying bacteria in soil are heterotrophic, and, as such, require a source of electrons or reducing equivalents contained in C compounds of organic matter or plant residues. Availability of organic C may thus affect both decomposing and denitrifying soil microorganisms.

In most reported studies on decomposition and $N_2O$ emissions, dried and often ground plant material was used. This facilitates a homogenous distribution in soil and minimizes differences between replicates. Nevertheless, drying of fine roots

prior to incubation increased their decomposition rate and led to overestimation of decomposition and nutrient cycling rates (Ludovici and Kress, 2006). Additionally, formation of plant litter associated anaerobic hotspots was reduced when ground plant material was homogenously mixed with the soil, while litter aggregation significantly increased soil $N_2O$ emissions (Loecke and Robertson, 2009). Differences in $N_2O$ emissions between two clover species were observed only with intact (but dried) leaves, but not when ground material was used (Kravchenko et al., 2018).

The aim of this study was 1) to evaluate how fresh maize root litter affects $N_2O$ emissions compared to fresh maize shoot litter, 2) to assess to what extend $N_2O$ emissions are related to the interaction of C and N mineralization from soil and litter, and 3) to analyze the changes in soil microbial community structures related to litter input and $N_2O$ emissions. We hypothesize that easily degradable C compounds stimulate microbial respiration in plant litter associated hotspots leading to high $N_2O$ formation when both C and N availability is high. We further expect that differences in litter chemical quality are

reflected in the structural composition of the soil microbial community.

Maize plants were grown in a greenhouse to produce root and shoot litter. To realize differences in soil $N_{min}$ concentration at harvest, two N fertilizer regimes (low vs. high) were applied. We then set up a laboratory incubation experiment with fresh maize root or root and shoot litter under fully controlled conditions and determined hourly $CO_2$ and $N_2O$ fluxes for 22 days. Soil samples were taken in regular intervals and analyzed for soil mineral N and water-extractable organic C (WEOC)

concentrations. At the end of the incubation experiment, soil microbial community structures were analyzed to identify adaptions to litter input.

## 2 Material and Methods

### 2.1 Preparation of plants and soils prior to incubation experiment

The soil for the experiment was collected 10 km south of Göttingen, Germany at the experimental farm Reinshof of the

University of Göttingen (51.484°N, 9.923°E). Soil was classified as Gleyic Fluvisol (21 % clay, 68 % silt, 11 % sand) containing 1.5 % C and 2.81 % humus, with a pH ($CaCl_2$) = 7.44.

Prior to the incubation experiment, maize plants were cultivated to obtain shoot and root biomass. For maize cultivation, Mitscherlich pots were filled with 5 kg air dried and sieved (2 mm) soil previously mixed with fertilizers (0.2 g N $kg^{-1}$ as $NH_4NO_3$, 0.14 g P $kg^{-1}$ as $Ca(H_2PO_4)_2$, 0.2 g K $kg^{-1}$ as $K_2SO_4$ and 0.04 g Mg $kg^{-1}$ as $MgSO_4 * 7 H_2O$ including

0.135 g S $kg^{-1}$). Soil moisture was adjusted to 25 Vol. % and volumetric water content (VWC) sensors (EC-5, Decagon Devices, Pullman, USA) were used to monitor soil water content. Six maize plants (*Zea mays* L. var. Ronaldinio) were sown





per pot and cultivated in a greenhouse with 16 h light and 8 h dark cycles. Pots were randomized in regular intervals to avoid microclimatic effects in the greenhouse.

To get different soil mineral N concentrations in soil, a second N fertilizer dose (0.2 g N kg$^{-1}$ as Ca(NO$_3$)$_2$ * 4 H$_2$O) was

applied to half of the pots six weeks after sowing. Soil with one N dose is referred to as N1 (0.2 g N kg$^{-1}$) and soil with two N doses is referred to as N2 (2 x 0.2 g N kg$^{-1}$). Plants were harvested 8 weeks after sowing: Maize plants were cut above the soil surface and roots were removed from soil by sieving and handpicking. Fresh roots were shaken and slightly brushed to remove adhering soil.

A subsample of aboveground maize biomass and maize roots was dried at 60°C to determine dry matter contents. To

determine water-extractable C and N concentrations, subsamples were extracted with H$_2$O$_{bidest}$ (maize root 1:1000 w/v, maize shoot 1:10000 w/v) for 16h and analyzed using a multi N/C® Analyzer (Model 3100, Analytik Jena, Jena, Germany). Another subsample was analyzed for the sum of structural components following established feedstuff analysis protocols based on the method proposed by Goering and Van Soest (1970), namely ash free neutral detergent fiber aNDFom, (VDLUFA, 2012a) acid detergent fiber ADFom, (VDLUFA, 2011) and acid detergent lignin ADL, (VDLUFA, 2012b).

According to the definitions, hemicellulose, cellulose and lignin contents were calculated as following: Hemicellulose = aNDFom - ADFom; Cellulose = ADFom - ADL; Lignin = ADL. Another subsample was milled using a ball mill and total carbon and nitrogen concentrations were analyzed using a C/N analyzer (Model 1110, Carlo Erba, Milano, Italy).

## 2.2 Incubation experiment

For the incubation experiment, shoot litter was taken from N2 plants only, and cut to a size of 2 cm. Roots of all plants (N1+N2) were mixed and also cut to 2 cm. Within each N level, all soil was mixed to ensure homogeneous initial conditions. Subsamples of both soils were taken for analysis of mineral N, water extractable C$_{org}$ concentration, and total soil C. C remaining from rhizodeposition, root hairs and small root fragments was calculated as the difference in soil C concentration before and after maize growth. The incubation experiment consisted of a two-factorial setup comprising two N

levels (N1 and N2) and three litter levels (Control = Cn, Root = Rt, Root+Shoot = RS) (see Table 1 and Figure 1 for details). Each treatment was replicated four times. Soil mineral N concentrations were 0.93 and 1.97 mg N kg$^{-1}$ for N1 and N2, respectively. Control soils (N1-Cn and N2-Cn) did not receive plant biomass, yet they contained C input from rhizodeposition of the previous maize growth. For the root treatment, 100 g fresh root biomass was added per kg dry soil (N1-Rt and N2-Rt), and in the root and shoot treatment, 100 g fresh root and 100 g fresh shoot biomass was added per kg dry

soil (N1-RS, N2-RS). Plant litter was homogeneously mixed with the soil, simulating residue incorporation and tillage.

PVC pots with a diameter of 20 cm and a total volume of 6.8 L were filled with fresh soil equivalent to 3.5 kg dry weight previously mixed with plant litter. Soil was compacted in a stepwise mode by filling a 2cm-layer of soil in pots and compacting it with a plunger. To ensure continuity between soil layers, the surface of the compacted layer was gently



scratched before adding the next soil layer. Due to high litter input, target bulk density was 1.1 g cm$^{-3}$. Actual bulk density
was determined by measuring headspace height, and these values were used for calculations.

To adjust soil moisture of all pots to 70% WHC, equivalent to 49% WFPS, water was dripped on the soil surface through
hollow needles (outer diameter 0.9 mm). Pots were covered with lids to minimize evaporation from the soil surface. The
incubation experiment was carried out under controlled temperature and light conditions (16 h day at 25°C, 8 h night at
19°C) for 22 days. Volumetric water content (VWC) sensors (EC-5, Decagon Devices, Pullman, USA) were used to monitor
soil water content.

**2.3 Gas sampling and analysis**

Gas fluxes were measured using the closed chamber method (Hutchinson and Mosier, 1981). Gas samples were taken every
12 hours (morning and evening) for the first 15 days and every 24 hours (midday) for the remaining 7 days. Due to technical
issues, gas samples taken in the morning of day 10 to day 15 had to be discarded. Before gas sampling, all pots were opened
140  for ventilation to ensure homogenous ambient air background conditions. Pots were closed with gastight PVC lids and 30 ml
gas samples were taken from each pot 0, 20, and 40 minutes after closure and filled into pre-evacuated 12 ml Exetainer glass
bottles (Labco, High Wycombe, UK). Samples were analyzed on a Bruker gas chromatograph (456-GC, Bruker, Billerica,
USA) deploying an electron capture detector (ECD) for $N_2O$ and a thermal conductivity detector (TCD) for $CO_2$. Samples
were introduced using a Gilson Autosampler (Gilson Inc., Middleton, WI, USA). Data processing was performed using
145  CompassCDS software. The analytical precision was determined by repeated measurements of standard gases (2500 and 550
ppm $CO_2$, 307, 760, and 6110 ppm $N_2O$) and was consistently < 2 %.

**2.4 Soil analyses**

Soil samples were taken from the pots using a soil auger of 16 mm diameter on 5, 9, 14 and 22 DAO (days after onset of
experiment). Holes were closed with glass tubes to avoid variation in the soil surface. Fresh subsamples were analyzed for
150  water extractable $C_{org}$ concentration (WEOC), and a subsample was frozen at -20°C for soil mineral N analysis.

Total soil carbon and nitrogen concentrations were analyzed using a C/N analyzer (Model 1110, Carlo Erba, Milano, Italy).
For determination of soil mineral N content, frozen samples were extracted with a 0.0125 M $CaCl_2$ solution (1:5 w/v) for
60 min on an overhead shaker (85 rpm). The extracts were filtered with 615 ¼ filter paper (Macherey - Nagel GmbH & Co.
KG, Düren, Germany) and stored at -20°C. The extracts were analyzed colorimetrically for the concentrations of $NO_3^-$ and
$NH_4^+$ using the San++Continuous-Flow Analyzer (Skalar Analytical B.V., Breda, The Netherlands). Soil water content was
determined with a parallel set of samples. Net N mineralization was calculated as the difference between the $NH_4^+$-N + $NO_3^-$-
N concentrations at the start and end of the incubation period plus N lost as $N_2O$ (Eq. 1).

$$Net\ mineralization = (NO_3^- + NH_4^+)_{end} - (NO_3^- + NH_4^+)_{start} + N_2O - N \qquad\qquad (1)$$





WEOC was determined according to Chantigny et al. (2007). Briefly, fresh soil was homogenized with deionized water
(1:2 w/v), samples were centrifuged and filtered with 0.45 µm polyether sulfone syringe filters (Labsolute, Renningen, Germany) and stored at -20°C. The extracts were analyzed using a multi N/C® Analyzer (Analytik Jena, Jena, Germany).

## 2.5 Analysis of bacterial community structures

### 2.5.1 DNA isolation and 16S rRNA gene amplification

To analyze the soil inhabiting bacterial communities, DNA was extracted from 0.5 g (fresh weight) soil sample taken at the
end of the incubation experiment (22 DAO) using the DNA extraction protocol described by Griffiths et al. (2000). Plant litter was removed from samples prior to extraction. In brief, cells were mechanically disrupted using bead beating and nucleic acids were extracted using phenol:chloroform:isoamyl alcohol (25:24:1; Carl Roth, Karlsruhe, Germany). Nucleic acids were then precipitated using polyethylene glycol (Carl Roth, Karlsruhe, Germany) and washed with 70% ice-cold ethanol (VWR, Radnor, Pennsylvania, USA). Subsequently, RNA was removed by RNase A digestion (Thermo Fischer
Scientific, Waltham, Massachusetts, USA) as described by the manufacturer. The RNA-free DNA was used for amplification of the V3 to V4 region of the 16S rRNA gene. We used the bacterial primer pair S-D-Bact-0341-b-S-17 and S-D-Bact-0785-a-A-21 targeting the V3-V4 region of the 16S rRNA gene described by Klindworth et al. (2013) with adapters for Illumina MiSeq sequencing. The PCR reaction mixture contained 5-fold Phusion GC buffer, 200 µM of each of the four deoxynucleoside triphosphates, 5% DMSO, 0.4 µM of each primer, 1 U of Phusion HF DNA polymerase (Fisher Scientific
GmbH, Schwerte, Germany), and 25 ng of RNA-free DNA as template. The following cycling scheme was used for DNA amplification: initial denaturation at 98 °C for 5 min and 25 cycles of denaturation at 98 °C for 45 s, annealing at 60 °C for 30 s and extension at 72 °C for 30 s, followed by a final extension at 72 °C for 10 min. For each sample, PCR reactions were performed in triplicate. Resulting PCR products were pooled in equimolar amounts and purified using the QIAquick Gel Extraction kit (Qiagen, Hilden, Germany) as recommended by the manufacturer. Quantification of the PCR products was
performed using the Quant-iT dsDNA HS assay kit and a Qubit fluorometer as described by the manufacturer (Invitrogen GmbH, Karlsruhe, Germany). Indexing of the PCR products was performed by the Göttingen Genomics Lab (G2L, Göttingen, Germany) using the Nextera XT Index kit as recommended by the supplier (Illumina, San Diego, CA, USA) and sequencing of 16S rRNA amplicons was performed using the dual index paired-end approach (2 × 300 bp) with v3 chemistry for the Illumina MiSeq platform.

### 2.5.2 Sequence processing

All bioinformatic processing of sequence data was done using Linux based software packages. Adapter removal and quality filtering of raw paired-end sequences was done using fastp v0.19.6 (Chen et al., 2018), with base correction in overlapped regions, a qualified quality phred of 20, size exclusion of sequences shorter than 50 bp and per read trimming by quality (phred 20). Merging of quality filtered paired-end reads was done by PEAR v0.9.11 (64 bit) with default parameters (Zhang





et al., 2014). Primer removal was conducted using cutadapt v1.18 (Martin, 2013). Subsequently, dereplication, denoising, as well as chimera detection and removal (denovo followed by reference based against the SILVA 132 SSU database), was performed with VSEARCH v2.13.0 (64 bit) (Rognes et al., 2016). Taxonomic classification of the amplicons sequence variants (ASVs, 100% sequence identity) was performed with BLAST+ v2.7.1 against the SILVA 132 SSU reference database (Quast et al., 2013). Subsequently, extrinsic domain ASVs and chloroplasts were removed from the dataset. Sample

comparisons were performed at the same surveying effort of 50.000 sequences. Statistical analyses were done using ASVs in R version 3.5.3 (R Core Team, 2019). All graphs were prepared using the R package *ampvis2* v2.4.7 (Andersen et al., 2018). Species richness, alpha diversity estimates, and rarefaction curves were determined using *ampvis2* v2.4.7 as well. To visualize the multivariate constrained dispersion Canonical Correspondence Analysis (CCA) was conducted with Hellinger transformed data (Legendre and Gallagher, 2001), ASV's with a relative abundance lower than 0.1% in any sample were

removed. Correlations of environmental parameters to the bacterial communities were analyzed using the *envfit* function of the *vegan* package v2.5-4 (Oksanen et al., 2015) and projected into the ordination with arrows with a p-value cutoff of 0.005. For further statistical analysis of the microbial community composition (on phyla, order and genus level) and diversity (Shannon, Simpson and PD index) multivariate generalized linear models (MGLM; with N level and litter addition as factors) as implemented in the *mvabund* R package v4.0.1 were employed with adjusted p-values (Wang et al., 2019). For

the generalized linear model analysis of variance (MGLM-ANOVA) tests, p-values < 0.05 were considered to be significant. In addition, we attempted to analyze the soil inhabiting fungal community using the fungal specific primer set ITS3_KYO2 and ITS4 (Toju et al., 2012), but were not able to amplify them.

**2.6 Calculations and statistical analyses**

All statistical analyses were performed using the statistical software R version 3.5.2 (R Core Team, 2018). Arithmetic means and standard error of the four replicates were calculated for hourly $CO_2$ and $N_2O$ fluxes. Cumulative gas emissions were calculated by linear interpolation between measured fluxes. To account for different C input in treatments, cumulative $CO_2$ and $N_2O$ emissions were standardized against the C input per treatment (see Table 1 for details on C input). Tukey's HSD test was used after analysis of variance to test for treatment effects (i.e., N level and litter addition) on cumulative $CO_2$

emissions. An interaction was identified between N level and litter addition on cumulative $N_2O$ emissions using interaction plots from the package *HH* v3.1-35 (Heiberger, 2018). A linear model using generalized least squares (gls) was fitted between cumulative $N_2O$ as response variable and N level, litter addition, and their interaction as fixed effects. Additionally, the model was fitted to account for inhomogeneous within-class variances. Estimated marginal means were then computed to analyze treatment effects using the R package *emmeans* v1.3.4 (Lenth, 2018). Several regression models were tested to

analyze the effect of maize litter on cumulative $N_2O$ emissions including the factors cumulative $CO_2$ emissions, initial soil $NO_3^-$ concentration, and net N mineralization during incubation period. For cumulative $CO_2$ emissions, regression models



included the factors total C input from litter, water-extractable C input from litter, hemicellulose fraction from litter input, cellulose fraction from litter input, and lignin fraction from litter input.

To evaluate effects of soil environmental variables on hourly $N_2O$ and $CO_2$ fluxes, a linear mixed effect model (lme) was fitted between $N_2O$ fluxes (ln transformed), soil $NO_3^--N$ and WEOC concentrations using the *lme* function from the package *nlme* v3.1-131 (Pinheiro et al., 2017). Pseudo-$R^2$ for lme was calculated using *r.squaredGLMM* from the package *MuMIn* v1.42.1 (Barton, 2018). Soil $NO_3^--N$ and WEOC concentrations between sampling dates were estimated by linear interpolation. Only evening/midday gas measurements were included in model calculations. To account for repeated measurements, incubation vessel and sampling day were set as random effects. Models were compared using maximum likelihood (ML), selected using AIC (Akaike's information criterion), and fitted using restricted maximum likelihood (REML).

All plots were made using the statistical software R version 3.5.2 (R Core Team, 2018) including the packages *plotrix* v3.7.4 (Lemon, 2006), *plot3D* v1.1.1 (Soetaert, 2017), and *viridisLite* v0.3.0 (Garnier, 2018).

## 3 Results

### 3.1 Chemical analyses of maize litter

Maize root and shoot litter differed in their chemical compositions (Table 2). Dry matter content of maize roots was much higher compared to shoot as roots had not been washed prior to analyses, so some soil adhering to roots was included in dry matter determinations. Thus, we calculated water-extractable concentrations in relation to total C instead of dry matter. Maize shoot litter was characterized by higher concentrations of water soluble C and N, and a higher share of easily degradable compounds like hemicellulose and cellulose compared to maize roots.

### 3.2 Hourly and cumulative $CO_2$ and $N_2O$ fluxes

Addition of maize litter increased hourly $CO_2$ fluxes compared to Control (Fig. 2), where addition of root and shoot litter (N1-RS, N2-RS) resulted in much higher fluxes compared to roots only (N1-Rt, N2-Rt). While absolute emission rates were strongly affected by litter input, time courses were similar in all litter treatments without visible differences between N1 and N2. $CO_2$ fluxes stayed on a similar level for the first ten days after onset of incubation showing fluctuations between morning and evening sampling times, and then constantly decreased until the end of the experiment.

After a short lag phase right after onset of experiment, hourly $N_2O$ emissions increased in all litter treatments compared to control treatments (Fig. 3 a+b). Highest hourly fluxes were measured in N2-RS, reaching 28 µg $N_2O$-N $kg^{-1}$ $h^{-1}$ on day 5. Fluxes stayed on a similar level from day 7 to day 15, and declined until the end of the experiment. Hourly $N_2O$ fluxes from root (N1-Rt, N2-Rt) and control treatments (N1-Cn, N2-Cn) remained on a low level during the whole incubation period ($\leq$ 1.11 µg and $\leq$ 0.1 µg $N_2O$-N$kg^{-1}$ $h^{-1}$, for -Rt and -Cn respectively). $N_2O$ fluxes from N1 were slightly lower than from N2 in





both litter treatments. Over all treatments and sampling dates, hourly $CO_2$ and $N_2O$ fluxes were positively correlated ($R^2=0.5993$, $p<0.001$, data not shown).

To account for different C inputs in treatments, cumulative $CO_2$ and $N_2O$ emissions were standardized against the C input

per treatment (Table 1). Still, cumulative $CO_2$ emissions were almost twice as high in -Rt and about four times higher in -RS compared to -Cn, indicating that differences between litter treatments cannot simply be explained by differences in C input. Addition of maize root and shoot litter increased cumulative $N_2O$ emissions by roughly 100-times compared to control treatments. In contrast, root litter increased cumulative $N_2O$ emissions only by a factor of 5.4 (N1-Rt) and 7 (N2-Rt) compared to the respective controls.

**3.3 Soil $NO_3^-$, $NH_4^+$ and water-extractable $C_{org}$ concentrations**

Addition of maize litter affected the time course of soil $NO_3^-$, $NH_4^+$ and WEOC concentrations (Fig. 4 a-c). In control treatments, initial soil $NO_3^-$ concentrations of 0.93 (N1-Cn) and 1.97 mg $NO_3^-$-N kg$^{-1}$ dry soil (N2-Cn) continuously increased until the end of the experiment reaching concentrations of 8.24 m N kg$^{-1}$ (N1-Cn) and 11.74 mg N kg$^{-1}$ (N2-Cn) respectively. Soil $NH_4^+$ concentrations showed variations on a low level only. Soil $NO_3^-$ concentrations were continuously

higher in N2 than in N1 and differences in soil $NH_4^+$ concentration were small. Higher fertilization in N2 during previous plant growth led to higher residual organic N and higher net N mineralization (7.61 and 10.08 mg N kg$^{-1}$ for N1-Cn and N2-Cn, respectively) during the incubation experiment. In treatments with litter, soil $NO_3^-$ concentrations decreased after an initial increase. In root treatments, soil $NO_3^-$ concentrations continuously decreased until the end of the incubation experiment to 1.9 (N1-Rt) and 2.5 mg N kg$^{-1}$ (N2-Rt), while in root plus shoot treatments soil $NO_3^-$ concentrations increased

again until the end of the experiment, reaching concentrations of 9.46 (N1-RS) and 9.52 mg N kg$^{-1}$ (N2-RS). During the whole incubation period, soil $NO_3^-$ concentrations in -RS were higher than in -Rt. Soil $NH_4^+$ concentrations only marginally increased for -Rt. Contrary to -Rt and -Cn, soil $NH_4^+$ concentrations increased until the end of the incubation experiment to 1.68 (N1-RS) and 1.52 mg N kg$^{-1}$ (N2-RS) in root and shoot treatments. Net N mineralization was 1.44 (N1-Rt) and 1.10 mg N kg$^{-1}$ (N2-Rt) in root treatments, and 14.32 (N1-RS) and 14.14 mg N kg$^{-1}$ (N2-RS) in root and shoot treatments. Maize

root litter did not affect WEOC, as concentrations were similar to Control throughout the incubation period. However in -RS treatments, WEOC increased after onset of incubation, reaching highest values (45.32 mg C kg$^{-1}$) for N1-RS at day 9, after which they decreased until the end of the experiment.

**3.4 Relations between $N_2O$ emissions and C and N parameters of plant litter and soil**

To identify the effect of N and C availability on hourly $N_2O$ fluxes, a linear mixed effect model was applied. The best model

included a significant interaction between soil $NO_3^-$ and WEOC ($p<0.0394$, Pseudo-$R^2=0.8888$, Table 4), and incubation vessel and sampling time as random parameters. Predictions of $N_2O$ fluxes based on this model are shown in (Fig. 5).

Linear regression analyses were used to identify relations between cumulative $CO_2$ and $N_2O$ emissions, litter quality and N parameters. Either hemicellulose + cellulose fraction or water-extractable C fraction of plant litter explained more than 96%





of variance of total cumulative $CO_2$ emissions ($p<2.2e^{-16}$) (Table 5). Regression analyses of the relationships between total

cumulative $N_2O$ emissions and influencing factors identified a strong positive relationship between total cumulative $N_2O$ emissions and total cumulative $CO_2$ emissions ($R^2=0.9362$, $p<7.632\ e^{-15}$) (Table 6), and between cumulative $N_2O$ emissions and mineralized N ($R^2=0.5791$, $p<9.551\ e^{-06}$), while initial soil $NO_3^-$ concentration did not explain any variance.

### 3.5 Bacterial community structure

The comparison over all maize litter treatments revealed that the bacterial diversity was slightly higher in N1 than in N2 soil

as shown by a higher number of amplicon sequence variants (ASVs, $R^2=0.1195$, $p=0.059$, Fig. S1). In addition, the alpha diversity indices Shannon ($R^2=0.1844$, $p=0.023$) and Simpson ($R^2=0.1131$, $p=0.065$), as well as Faith's phylogenetic diversity (PD; $R^2=0.1844$, $p=0.059$) were higher for N1 than for N2 samples (Table S4).

The canonical correspondence analysis revealed a significant correlation ($p < 0.001$) of the bacterial community composition with total $CO_2$ ($R^2 = 0.6758$) and $N_2O$ ($R^2 = 0.6179$) emissions, and the litter level, expressed by a clear separation of the N1-

RS and N2-RS samples of all other samples (Fig. 6). With increasing C input, N2 samples cluster more closely than N1 samples. No significant correlation of litter level and microbial diversity was observed, PD index increased in N1 samples with increasing C input, while the opposite was found for N2 samples. Comparison of N1-C and N1-RS revealed no difference in diversity indices (Shannon and Simpson), while N1-R showed lower Shannon and Simpson diversity indices (Table S4). The Shannon diversity index was lowest in N2-R comparing all N2 treatments, while the Simpson index was

lowest for N2-RS.

Overall, the soil bacterial communities were dominated by *Actinobacteria*, *Proteobacteria* and *Chloroflexi* accounting for 15 to 31% (Fig. S2). The highest relative abundance of *Actinobacteria* and *Chloroflexi* was found in N2-R and of *Proteobacteria* in N1-R. Among these phyla, the order *Gaiellales* (*Actinobacteria*), *Sphingomonadales* (*Proteobacteria*) and *Thermomicrobiales* (*Chloroflexi*) showed the highest relative abundance, especially in N2-R (9.3%), N1-R (7.5%) and N2-

RS (9%), respectively. Nevertheless, the phyla *Acidobacteria*, *Planctomycetes*, *Verrucomicrobia*, *Gemmatimonadetes*, *Firmicutes*, *Patescibacteria* and *Bacteroidetes* were also detected (>1%) (Fig. 7). In detail, *Bacteroidetes* and *Gemmatimondadetes* decreased (with a negative slope, but not significant) with increasing N level, while the abundance of *Firmicutes* increased significantly ($p=0.038$). In addition, although present only in low relative abundance, the *Cyanobacteria* decreased significantly ($p=0.003$) with increasing N levels. At the genus level, *Pseudomonas*,

*Altererythrobacter*, *Gaiella*, *Nocardioides*, *Agromyces*, *Bacillus*, and *Lysobacter* were most abundant accounting for up to 5.7 % of all ASVs. The genera *Bacillus*, *Gaiella*, *Altererythrobacter*, *Blastococcus*, and *Pseudomonas* showed highest abundance in N2 samples, while *Lysobacter*, and *Sphingomonas* were more abundant in N1 samples (Fig. S3). The most abundant classified species found were *Agromyces sp.*, *Bacillus sp.* and *Sphingomonas sp.* Nevertheless, species such as *Pseudomonas sp.*, *Nitrosospira sp.*, *Nitrosospira briensis*, *Alcaligenes sp.* and *Mesorhizobium sp.* were also identified.

Overall, the bacterial community composition was significantly influenced by N-level ($p=0.005$) and maize litter treatment ($p=0.033$).



## 4 Discussion

### 4.1 Decomposability of maize litter

Maize root and shoot litter quality controlled $NO_3^-$ and WEOC availability and decomposition related $CO_2$ emissions during

the initial phase of maize litter decomposition. Harvest of plants, removal of roots and mixing of soil fostered mineralization and nitrification, as reflected by gradually increasing soil $NO_3^-$ concentrations. The absence of changes in soil $NH_4^+$ concentrations in control treatments without litter addition (N1-Cn, N2-Cn) indicate that all $NH_4^+$ was directly nitrified. Also in controls, available C was low as indicated by low $CO_2$ emissions and decreasing WEOC concentrations.

The potential for mineralization in soil is known to be high after tillage (Höper, 2002) and positive net mineralization has

been reported in control soil without litter addition (Machinet et al., 2009; Velthof et al., 2002), and in the fallow period after rice harvest (Aulakh et al., 2001).

Maize shoot litter was characterized by a high share of easily degradable compounds. High percentages of water-soluble N and water-soluble $C_{org}$ from maize shoot litter strongly increased soil WEOC and $NO_3^-$ concentrations. Availability of easily degradable compounds was also reflected by strongly increased hourly and cumulative $CO_2$ fluxes from N1-RS and N2-RS.

While net mineralization in -RS was similar to -Cn, it was very small in -Rt indicating that N from mineralization was immobilized by soil microorganisms to decompose root C compounds (Robertson and Groffman, 2015). Cumulative $CO_2$ emissions in litter treatments were clearly higher than in control, but hourly $CO_2$ fluxes continuously decreased after onset of incubation, as easily degradable C was consumed. This is in accordance with results of Hu et al. (2016), who reported that maize fine root input initially increased $CO_2$ fluxes, which then decreased during the first 20 days of incubation.

Mineralization of plant litter may increase soil $NO_3^-$ concentrations especially when C:N ratios are low (Li et al., 2013; Millar and Baggs, 2004). However, net N immobilization has been reported after addition of roots of maize (Machinet et al., 2009; Mary et al., 1993; Velthof et al., 2002), wheat (Jin et al., 2008; Velthof et al., 2002), barley and sugar beet (Velthof et al., 2002), reaching a maximum around day 21 (Mary et al., 1993). Chemical composition has been proven to be the primary controller of decomposition rates of both roots (Birouste et al., 2012; Redin et al., 2014; Silver and Miya, 2001) and

aboveground plant litter (Jensen et al., 2005; Zhang et al., 2008) of many different species. Slower decomposition of roots compared to leaves and stems was related to differences in chemical composition of plant organs (Jenkinson, 1965; Johnson et al., 2007). Accordingly, decomposition of roots from 16 maize genotypes was controlled by soluble residue components in the short term whereas lignin and the interconnections between cell wall polymers were important in the long-term (Machinet et al., 2011). In our study, regression analyses identified a strong positive relationship between cumulative $CO_2$

emissions and water-extractable C fraction of plant litter ($R^2$=0.966, $p< 2.2*e^{-16}$) (Table 5).

### 4.2 $N_2O$ emissions as affected by biodegradability of maize litter and soil N level

Denitrification in soil is largely controlled by the supply of readily decomposable organic matter (Azam et al., 2002; Burford and Bremner, 1975; Loecke and Robertson, 2009). Availability of easily degradable C compounds stimulates microbial




respiration, limiting $O_2$ at the microsite level and increasing $N_2O$ emissions (Azam et al., 2002; Chen et al., 2013; Miller et
al., 2008) leading to significant correlations between hourly and cumulative $N_2O$ and $CO_2$ emissions (Azam et al., 2002;
Fiedler et al., 2017; Frimpong and Baggs, 2010; Huang et al., 2004; Millar and Baggs, 2004, 2005). We identified a strong
positive correlation ($R^2$=0.9362, $p \leq 7.632$ e$^{-15}$) between cumulative $CO_2$ and $N_2O$ emissions (Table 6), supporting our
hypothesis that litter quality, especially quality of C compounds, affects $N_2O$ fluxes from denitrification by creating plant
litter associated microsites with low $O_2$ concentrations.

High mineralization in -RS treatments may have especially favored coupled nitrification-denitrification where $NO_2^-$ and
$NO_3^-$ are produced by nitrifiers in aerobic habitats and subsequently denitrified by denitrifiers in close-by anaerobic habitats
(Butterbach-Bahl et al., 2013; Wrage et al., 2001). Here, $N_2O$ is mainly produced in the interface of aerobic and anaerobic
zones, which are typically found in plant litter associated hotspots (Kravchenko et al., 2017). In addition, $N_2O$ can also be
produced aerobically during heterotrophic and autotrophic nitrification (Anderson et al., 1993; Van Groenigen et al., 2015;
Wrage et al., 2001; Zhang et al., 2015). In both processes, $N_2O$ can be formed as byproduct from chemical hydroxylamine
oxidation (Butterbach-Bahl et al., 2013; Van Groenigen et al., 2015). Nitrifier denitrification as a pathway of autotrophic
nitrification has been reported mostly under soil conditions differing from our study, namely high $NO_2^-$, $NH_3$ or urea
concentrations, and low organic C availability (Wrage-Mönnig et al., 2018; Wrage et al., 2001). In contrast, with high
availability of organic C and N compounds, high $N_2O$ emissions from heterotrophic nitrification have been reported
(Anderson et al., 1993; Hu et al., 2016; Papen et al., 1989; Wrage et al., 2001). Zhang et al. (2015) reported 72-77% of $N_2O$
being produced by heterotrophic nitrification with an arable soil under incubation conditions similar to our study. However,
Li et al. (2016) estimated that denitrification was the dominant source of $N_2O$ in residue-amended soil at 40-60 % WFPS.
High correlation of cumulative $N_2O$ emissions and mineralized N during the incubation period ($R^2$=0.5791, $p<9.551$ e$^{-06}$)
indicates that heterotrophic nitrification may have contributed to $N_2O$ production in our study. To differentiate further
between processes contributing to $N_2O$ production, stable isotope methods can be used (Baggs, 2008; Butterbach-Bahl et al.,
2013; Van Groenigen et al., 2015; Wrage-Mönnig et al., 2018).

Another aim of this study was to investigate the effect of residual mineral N on plant litter induced $N_2O$ emissions. To this
end, we included two N levels that were obtained by different N fertilization during the pre-experimental plant growth phase
(N1: 0.2 g N kg$^{-1}$, N2: 2 x 0.2 g N kg$^{-1}$). At the onset of the incubation experiment, soil mineral N concentration was twice as
high in N2 compared to N1, but generally very low (0.93 and 1.97 mg $NO_3^-$-N kg $^{-1}$ dry soil for N1 and N2, respectively).
Higher N fertilizer input in N2 during plant growth led to lower C input from rhizodeposition (Tab. 1), which is consistent
with literature findings (Kuzyakov and Domanski, 2000; Paterson and Sim, 1999). Cumulative $N_2O$ emissions were in
tendency higher in N2 than in N1, suggesting that $NO_3^-$ was limited, especially in -RS treatments where C availability was
highest. In addition, litter chemical quality strongly affected N availability. The interdependency of C and N availability was
validated by analyses of regression highlighting a strong positive interaction between soil $NO_3^-$ and WEOC concentrations
resulting in much higher $N_2O$ emissions only when both $NO_3^-$ and WEOC were available. This further supports our findings





that high bioavailability of maize shoot litter increased microbial respiration by heterotrophic microorganisms resulting in plant litter associated hotspots with high $N_2O$ formation.

Variation in $N_2O$ emissions is often related to quality parameters of plant residues, mostly the C:N ratio (Baggs et al., 2000; Chen et al., 2013; Millar and Baggs, 2004; Novoa and Tejeda, 2006). Especially easily degradable fractions, such as water-soluble C (Burford and Bremner, 1975) or the holocellulose fraction (hemicelluloses + cellulose) (Jensen et al., 2005), explained a large share of variability of C mineralization and $N_2O$ emissions, while lignin content was not relevant (Redin et al., 2014; Silver and Miya, 2001). Comparing 28 laboratory and field studies, Chen et al. (2013) reported that microbial growth-induced microsite anaerobicity could be the major driver for the dynamic change in soil $N_2O$ emissions following residue amendment and Kravchenko et al. (2017) showed that water absorption by plant residues further enhances formation of plant-litter associated anaerobic hotspots. In the initial phase of decomposition, water-soluble compounds (sugars, amino acids) are leached from litter providing easily degradable compounds for microbial metabolism. After litter addition, $CO_2$ fluxes increased immediately due to increased respiration, rapidly reducing $pO_2$, and creating anaerobic microsites. We anticipate that formation of such hotspots was further enhanced by the amount of litter addition, as litter input was higher in -RS than in -Rt, and higher compared to other studies (Chen et al., 2013).

### 4.3 Bacterial community response to maize litter input and soil N level

After litter addition, the bacterial community adapts within a few days to substrate availability (Pascault et al., 2013). The canonical correspondence analysis (CCA) showed a clear correlation of the soil inhabiting bacterial community, litter input and total $CO_2$ and $N_2O$ emissions. As shown by the CCA, the bacterial community structure in N1-RS and N2-RS was distinct from those in the control samples and soil with addition of root residues. Combined addition of root and shoot litter affected the soil bacterial community leading to a less diverse and more specialized community structure, which was also shown by the alpha diversity indices (see supplemental Table S1). A significant reduction of soil bacterial diversity was induced by different N levels, as previously shown by Zeng et al. (2016). In addition, Rousk and Bååth (2007) observed a negative correlation between mineral N addition and bacterial growth, while the addition of barley straw and alfalfa correlated positively. The phylogenetic diversity (PD) supports these findings by showing a more complex picture. While PD in N1 samples increased with increasing C input, it decreased in N2 samples with increasing C input, indicating a shift of the influencing factors from the C input to the N level.

The most abundant phyla in our soil samples were the *Actinobacteria*, *Proteobacteria* and *Chloroflexi*. Among these phyla, the genera *Pseudomonas* (*Proteobacteria*), *Gaiella* (*Actinobacteria*) and *Thermomicrobiales* (*Chloroflexi*) showed the highest abundance in the N2 samples, indicating their involvement in N-cycling. *Pseudomonas* species such as *Pseudomonas aeruginosa*, *P. stutzeri*, and *P. denitrificans* are known to reduce $NO_3^-$ and to contribute to $N_2O$ and $N_2$ emissions (Carlson and Ingraham, 1983). *Gaiella occulta*, belonging to the *Actinobacteria*, is also known for the reduction of $NO_3^-$ to $NO_2^-$ (Albuquerque et al., 2011). The genus *Thermomicrobiales* comprises species which can grow on nitrate, ammonia and





alanine as sole nitrogen sources and are able to hydrolyze cellulose or starch (Houghton et al., 2015). Relative abundance of

*Thermomicrobiales* increased with N and C input, indicating favorable growth conditions for this genus (Fig. 7).

We identified several genera involved in C cycling including members of *Agromyces*, *Bacillus*, and *Micromonospora*. *Agromyces ulmi* was present in low abundance in our samples and it is known to contribute to C cycling in soils through xylanolytic activity (Rivas et al., 2004). Members of the genus *Bacillus* (*Firmicutes*) have been reported to play a crucial role in carbon cycling in a wide range of environments by functions such as plant growth promotion or production of amylases

and cellulases (Lyngwi and Joshi, 2014). Among the genus *Bacillus*, we found one species, *Bacillus sp.* KSM-N252, in relatively high abundance (1-2%) in N2 samples. This species encodes an alkaline endoglucanase, which can hydrolyze cellulose (Endo et al., 2001). Similarly, *Micromonospora* (*Actinobacteria*) are known to produce hydrolytic enzymes showing cellulolytic and xylanolytic activity (Carro et al., 2018; de Menezes et al., 2012). Abundance of *Bacillus sp.* KSM-N252 (N2-C 2%, N2-R 1.1% and N2-RS 0.8%) and *Micromonospora* (N2-R 1.9%, N2-RS 1%) decreased with increasing

input of water-extractable C indicating that cellulose was only decomposed when no easily degradable C was available.

Culture-independent sequence techniques have revealed that members of the phyla *Actinobacteria*, *Chloroflexi*, *Firmicutes*, *Bacteroidetes* and *Nitrospirae* possess nirK or nirS genes, and can reduce nitrite to nitric oxide (Cantera and Stein, 2007; Nolan et al., 2009). In our treatments, *Actinobacteria*, *Chloroflexi*, and *Firmicutes* were more abundant in N2 samples, whereas *Bacteroidetes*, and *Nitrospirae* were more abundant in N1 samples which may indicate that the latter are more

competitive under conditions of very low mineral nitrogen availability in soil. The reduction of nitrate has been shown for *Mesorhizobium sp.* (Okada et al., 2005) and *Rhizobium sp.* (Daniel et al., 1982). Although only in low abundance, we found these species predominantly in N2 samples. Species belonging to the genus *Agromyces* (*Actinobacteria*) are also known to reduce nitrate (Zgurskaya et al., 2008). In addition, species capable of denitrification under anaerobic, $O_2$-limited and aerobic conditions can be found in the genera *Bacillus* and *Micromonospora*, as well as *Pseudomonas* and *Rhodococcus*

(Verbaendert et al., 2011) that were more abundant in N2 samples.

The higher abundances of C cycling and N reducing bacteria in N2 samples reflects the tendency of increased $N_2O$ emissions with increasing N level and further supports our hypothesis that C and N availability from plant litter were the main drivers of $N_2O$ emissions in our study.

## 5 Conclusions

We examined $CO_2$ and $N_2O$ emissions after simulated post-harvest incorporation of maize root or root plus shoot litter in a laboratory incubation study. High bioavailability of maize shoot litter strongly increased microbial respiration in plant litter associated hotspots leading to increased $N_2O$ emissions when both C and $NO_3^-$ were available. Coupled nitrification-denitrification and heterotrophic nitrification presumably contributed to $N_2O$ formation. Maize root litter was characterized by a higher share of slowly degradable C compounds and lower concentrations of water soluble N, hence formation of

anaerobic hotspots was limited and microbial N immobilization restricted $N_2O$ emissions. Bacterial community structures



reflected degradability of maize litter types. Its diversity decreased with increasing C and N availability, favoring fast growing C cycling and N reducing bacteria, namely *Actinobacteria*, *Chloroflexi*, *Firmicutes* and *Proteobacteria*.

**Data availability:** The 16S rRNA gene sequences were deposited in the National Centre for Biotechnology Information
(NCBI) Sequence Read Archive (SRA) under bioproject number PRJNA557843. Data from measurements are available upon request from the corresponding author.

**Supplement:** The supporting information related to this study will be published online.

**Author contributions:** PSR, RW and KD designed the experiments and PSR carried them out. BP and DS carried out
microbial analyses, sequence processing and provided figures. JP, RW and KD contributed to interpretation of results. PSR prepared the manuscript with contributions from all co-authors.

**Competing interests:** The authors declare no competing interests.

**Acknowledgments:** The authors thank Jakob Streuber, Simone Urstadt, and Finn Malinowski for gas sampling and laboratory analyses, as well as Alexander Silbersdorff (ZfS Statistical Consulting) and Oliver Caré for advise on data handling and statistical analysis.

This study was funded by the Deutsche Forschungsgemeinschaft through the research unit DFG-FOR 2337: Denitrification in Agricultural Soils: Integrated Control and Modelling at Various Scales (DASIM).

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





**Table 1: Two-factorial setup of the incubation experiment. Soil $N_{min}$ concentrations were measured directly before onset of the incubation experiment. C input in Control is from rhizodeposition (RD) only, C input in Root is from rhizodeposition and roots, C input in Root + Shoot is from rhizodeposition, roots and shoot biomass. N input is from root and shoot biomass, respectively.**


| N level | $N_{min}$ [mg $NO_3^-$-N kg$^{-1}$ dry soil] | Treatment | Biomass input [g FM kg$^{-1}$ dry soil] | C input [g C kg$^{-1}$ dry soil] | N input [g N kg$^{-1}$ dry soil] |
|---|---|---|---|---|---|
| N1 | 0.93 | Control | RD | 3.47 | n.d. |
| | | Root | RD + 100 | 3.47 + 4.18 = 7.65 | 0.25 |
| | | Root + Shoot | RD + 100 + 100 | 3.47 + 4.18 + 6.16 = 13.80 | 0.25 + 0.27 = 0.52 |
| N2 | 1.97 | Control | RD | 2.74 | n.d. |
| | | Root | RD + 100 | 2.74 + 4.18 = 6.92 | 0.25 |
| | | Root + Shoot | RD + 100 + 100 | 2.74 + 4.18 + 6.16 = 13.07 | 0.25 + 0.27 = 0.52 |





**Table 2: Chemical characteristics of maize root and shoot litter used in the incubation experiment. Hemicellulose and cellulose are expressed relative to lignin content.**

|  | Root | Shoot |
| --- | --- | --- |
| Dry matter [%] | 62.9 | 14.7 |
|  |  |  |
| C:N ratio | 17.0 | 23.2 |
| Lignin:N ratio | 2.82 | 1.44 |
| Water soluble $C_{org}$ [% of total C] | 11.6 | 23.4 |
| Water soluble N [% of total N] | 8.8 | 25.8 |
|  |  |  |
| Hemicellulose (relative content) | 3.36 | 9.08 |
| Cellulose (relative content) | 3.18 | 11.5 |
| Lignin (relative content) | 1 | 1 |






**Table 3: Absolute cumulative $N_2O$ and $CO_2$ emissions and relative to C input and $N_2O$ / $CO_2$ ratio of 22-day incubation experiment with two pre-incubation-N-levels (N1, N2) and three litter addition treatments (Control = no litter input, Root = 100 g root FM kg$^{-1}$ dry soil, Root+Shoot = 100 g root FM + 100g shoot FM kg$^{-1}$ dry soil).**

| N level | Treatment | $N_2O$ [µg $N_2O$-N kg$^{-1}$] | | $N_2O$ [µg $N_2O$-N kg$^{-1}$ g$^{-1}$ C input] | | $CO_2$ [mg $CO_2$-C kg$^{-1}$] | | $CO_2$ [mg $CO_2$-C kg$^{-1}$ g$^{-1}$ C input] | | $N_2O/CO_2$ ratio [µg N mg$^{-1}$ C] |
|---|---|---|---|---|---|---|---|---|---|---|
| N1 | Control | 10.21 ± 4.23 | a | 2.95 ± 1.22 | a | 141.89 ± 29.74 | a | 40.94 ± 8.58 | a | 0.07 |
| | Root | 120.91 ± 24.09 | b | 15.81 ± 3.15 | b | 533.51 ± 83.19 | b | 69.78 ± 10.88 | b | 0.23 |
| | Root+Shoot | 4337.31 ± 424.98 | c | 314.25 ± 30.95 | c | 2287.23 ± 289.48 | c | 165.72 ± 20.97 | c | 1.91 |
| | | | | | | | | | | |
| N2 | Control | 11.35 ± 6.75 | a | 4.15 ± 2.47 | a | 129.44 ± 47.47 | a | 47.30 ± 17.35 | a | 0.08 |
| | Root | 201.14 ± 105.62 | ab | 29.08 ± 15.27 | ab | 647.48 ± 196.13 | ab | 93.61 ± 28.36 | b | 0.31 |
| | Root+Shoot | 5357.87 ± 1193.50 | c | 409.82 ± 91.30 | c | 2361.19 ± 287.20 | c | 180.63 ± 21.97 | c | 2.25 |

Values represent means (n=4) ± standard deviation. Different letters in the same column indicate a significant difference according to the Tukey's HSD post-hoc tests at $p \leq 0.05$.

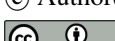



**Table 4: Significance of fixed effects of soil $NO_3^-$-N (mg $NO_3^-$-N $kg^{-1}$), water extractable organic C (WEOC, mg C $kg^{-1}$) and first-order interaction on $N_2O$ fluxes (µg $N_2O$-N $kg^{-1}$ $h^{-1}$; ln transformed) using linear mixed effect model.**

|  | Estimate | Standard error | p value |
| --- | --- | --- | --- |
| Intercept | -0.4237 | 0.2074 | 0.0416 |
| $NO_3^-$-N | 0.0152 | 0.0248 | 0.5401 |
| WEOC | 0.0210 | 0.0077 | 0.0065 |
| $NO_3^-$-N : WEOC | 0.0023 | 0.0011 | 0.0394 |





**Table 5: Results of regression analyses of the relationship between total cumulative $CO_2$ emissions and C quality parameters of plant litter (AICc = Akaike's information criterion).**

| Regression model | residual standard error | degrees of freedom | adjusted $R^2$ | p value | AICc |
|---|---|---|---|---|---|
| $CO_2$ ~ total litter C input | 274.5 | 22 | 0.9213 | $7.65*e^{-14}$ | 342.73 |
| $CO_2$ ~ water-soluble C input | 181.9 | 22 | 0.9655 | $< 2.2*e^{-16}$ | 322.98 |
| $CO_2$ ~ Hemicellulose | 272.4 | 22 | 0.9225 | $6.497*e^{-14}$ | 342.38 |
| $CO_2$ ~ Cellulose | 221.1 | 22 | 0.9489 | $6.478*e^{-16}$ | 332.35 |
| $CO_2$ ~ Lignin | 496.6 | 22 | 0.7425 | $3.873*e^{-08}$ | 371.19 |
| $CO_2$ ~ Hemicellulose + Cellulose | 180.2 | 21 | 0.9661 | $< 2.2*e^{-16}$ | 324.32 |





**Table 6: Results of regression analyses of the relationship between total cumulative N$_2$O emissions, total cumulative CO$_2$ emissions and N parameters of plant litter and soil (AIC = Akaike's information criterion).**

| Regression model | residual standard error | degrees of freedom | adjusted R$^2$ | p value | AICc |
|---|---|---|---|---|---|
| N$_2$O ~ CO$_2$ | 593.9 | 22 | 0.9366 | 7.073*e$^{-15}$ | 379.78 |
| N$_2$O ~ initial soil NO$_3^-$ | 2404 | 22 | -0.03885 | 0.7119 | 446.89 |
| N$_2$O ~ Mineralized N | 2191 | 22 | 0.5791 | 9.551*e$^{-06}$ | 425.21 |





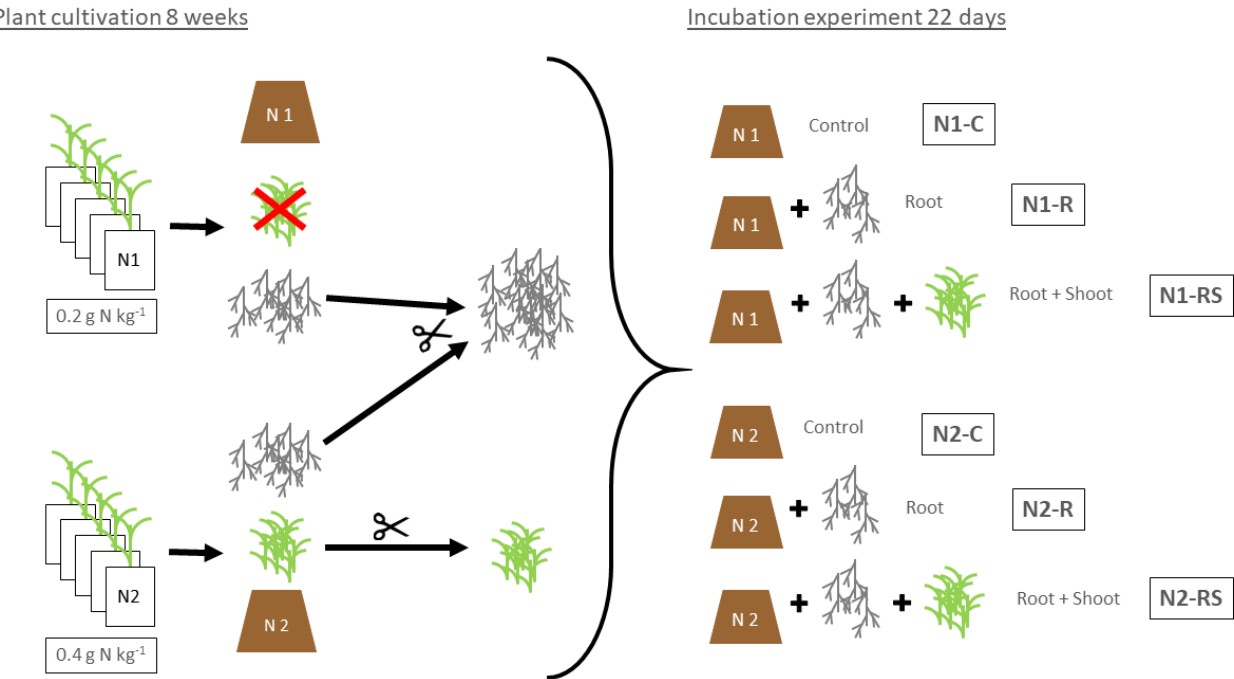

**Figure 1: Preparation and experimental setup of the incubation experiment. N1 (0.2 g N kg⁻¹) and N2 (2 x 0.2 g N kg⁻¹) referring to the N levels during plant growth. Control soil (N1-C and N2-C) without addition of plant biomass. Root treatment with addition of 100g fresh root biomass per kg dry soil (N1-R and N2-R) and Root + Shoot treatment with addition of 100 g root and 100 g shoot biomass per kg dry soil (N1-RS, N2-RS).**







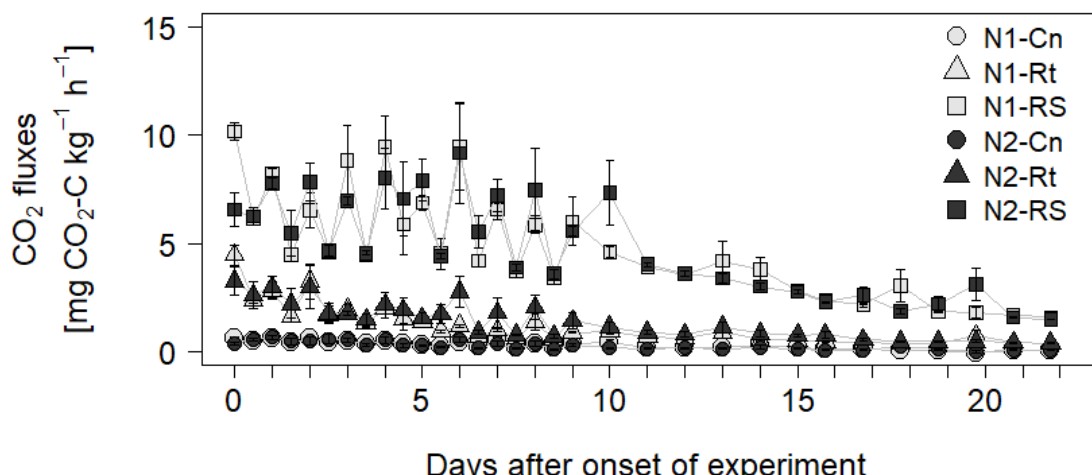

Figure 2: Hourly CO₂ emissions from soils with two N levels (N1, N2) after incorporation of maize root litter (-Rt), maize root + shoot litter (-RS) and control (-Cn) without litter. Error bars show standard error of mean values (n = 4). When not visible, error bars are smaller than the symbols.



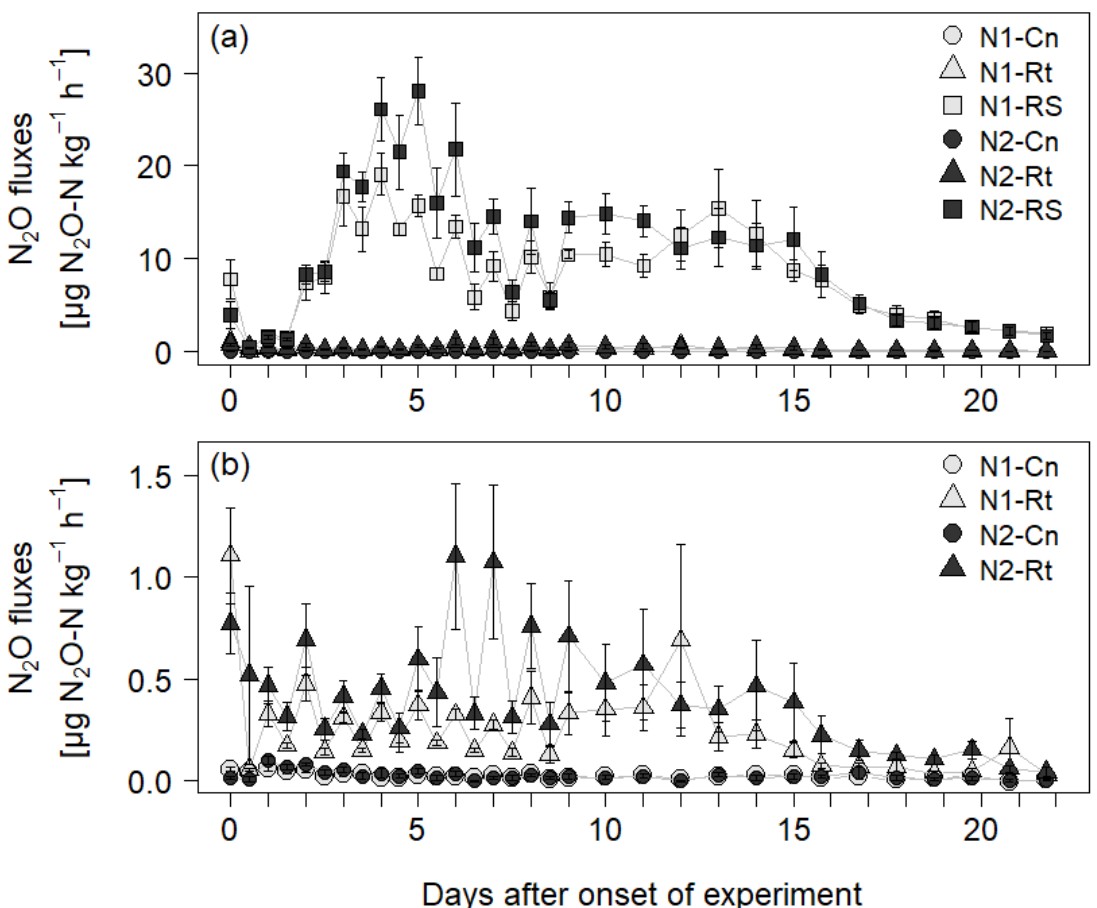

Figure 3 a+b: Hourly N$_2$O emissions from soils with two N levels (N1, N2) after incorporation of maize root litter (-Rt), maize root + shoot litter (-RS) and control (-Cn) without litter. Error bars show standard error of mean values (n = 4). When not visible, error bars are smaller than the symbols. Note: data of figure 3 b are excerpt from 3 a, and are shown with different scaling.


**Figure 4 a-c: NO$_3^-$, WEOC, and NH$_4^+$ concentration from soils with two N levels (N1, N2) after incorporation of maize root litter (-Rt), maize root + shoot litter (-RS) and control (-Cn) without litter. Error bars show standard error of mean values (n = 4) (day 0: n=3). When not visible, error bars are smaller than the symbols.**




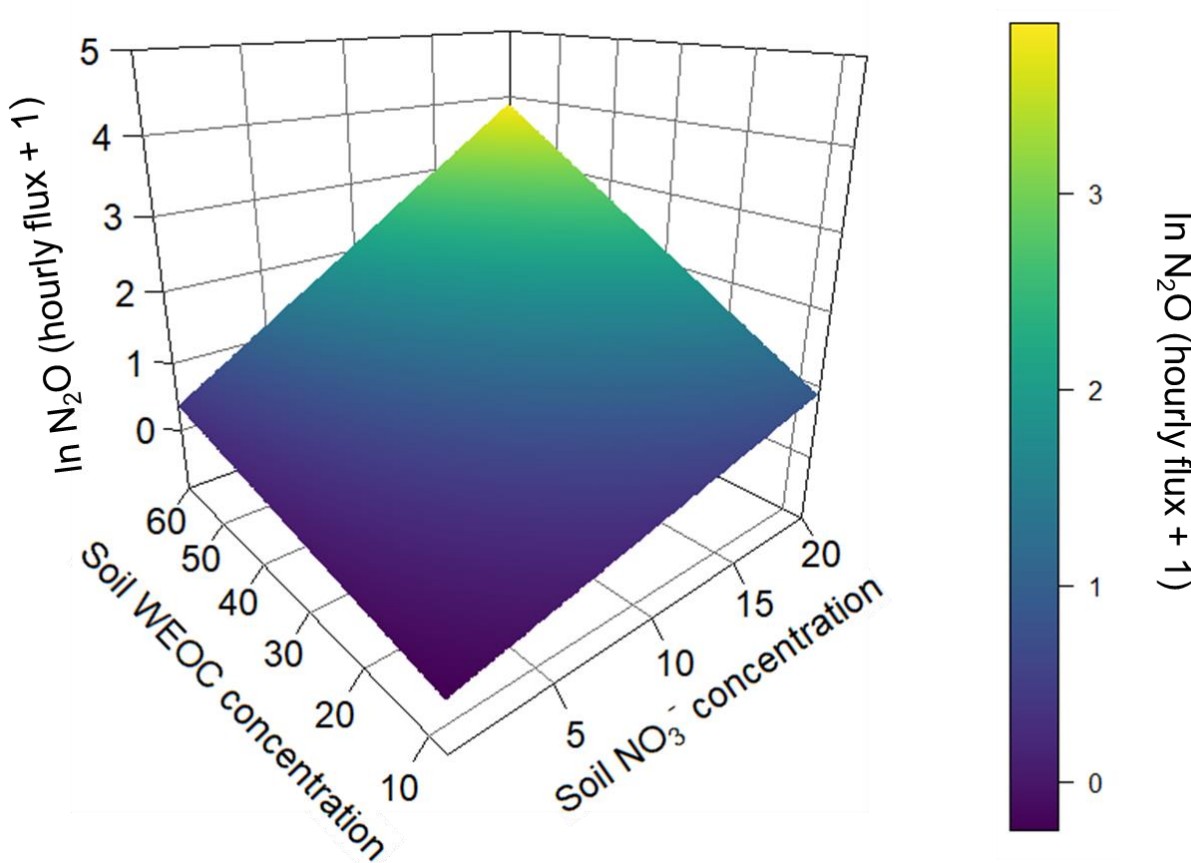


**Figure 5: Prediction of N₂O fluxes (µg N₂O-N kg⁻¹ h⁻¹) (ln transformed) based on soil NO₃⁻ and water extractable C_org concentrations based on linear mixed-effect model (Pseudo-R²=0.89).**





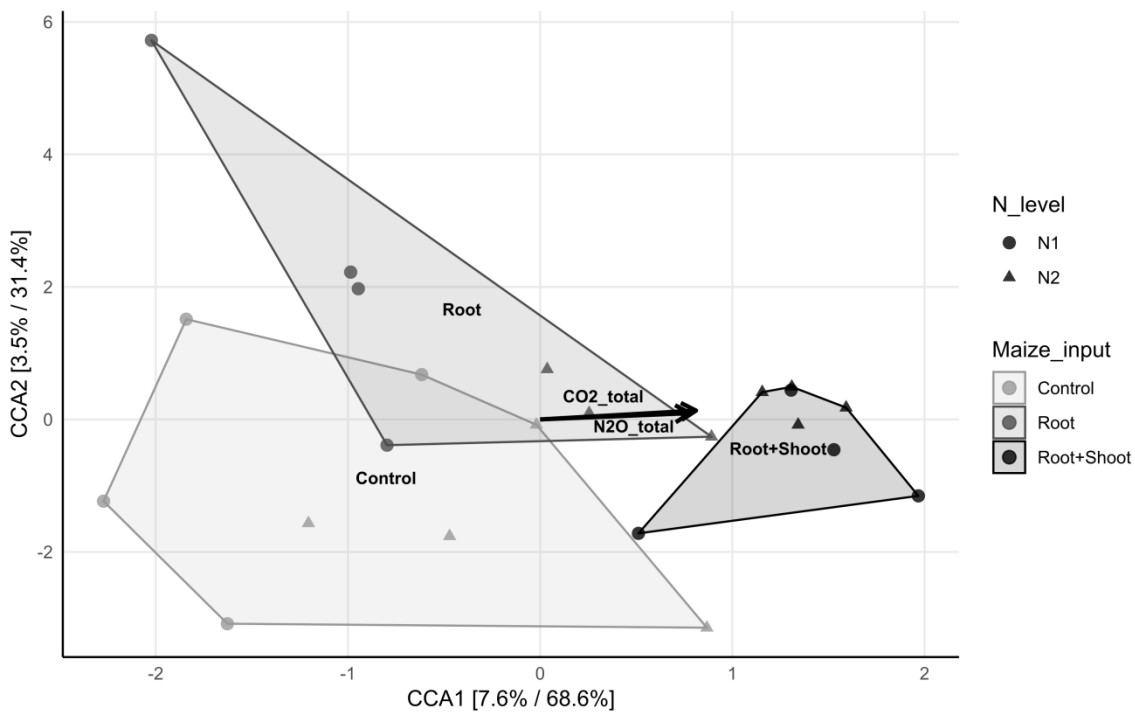


**Figure 6: Canonical Correspondence Analysis (CCA) displaying the compositional distribution of the soil inhabiting bacterial communities between the control (N1-C and N2-C; n=4), root (N1-R and N2-R; n=4 and n=3) and root + shoot treatment (N1-RS and N2-RS; n=4). Significant correlations of total $CO_2$ and $N_2O$ emissions are shown by black arrows (p ≤ 0.005). The relative contribution (eigenvalue) of each axis to the total inertia in the data as well as to the** 
**constrained space only, respectively, are indicated in percent at the axis titles.**



|  | N1 | | | N2 | | |
|---|---|---|---|---|---|---|
|  | Control | Root | Root+Shoot | Control | Root | Root+Shoot |
| Actinobacteria; Gaiellales | 6 | 7.2 | 6.3 | 7.4 | 9.4 | 7.5 |
| Actinobacteria; Micrococcales | 3 | 5.5 | 4.6 | 4.2 | 5 | 4.8 |
| Actinobacteria; Propionibacteriales | 2.6 | 3.9 | 3.3 | 3.1 | 3.7 | 2.9 |
| Actinobacteria; Solirubrobacterales | 2.1 | 2.5 | 2.4 | 2.7 | 3.2 | 3.1 |
| Actinobacteria; Frankiales | 1.4 | 2.1 | 1.8 | 1.9 | 2.4 | 3.9 |
| Actinobacteria; Microtrichales | 2.3 | 2.2 | 2.2 | 2.1 | 1.7 | 1.6 |
| Actinobacteria; Micromonosporales | 1 | 0.9 | 0.7 | 0.9 | 2.1 | 1.2 |
| Proteobacteria; Sphingomonadales | 5.9 | 7.5 | 5.2 | 6.6 | 5.9 | 5.7 |
| Proteobacteria; Pseudomonadales | 1.5 | 5.4 | 1.3 | 5.8 | 4.5 | 4.6 |
| Proteobacteria; Betaproteobacteriales | 3.7 | 4.8 | 2.4 | 3.2 | 1.9 | 2.8 |
| Proteobacteria; Xanthomonadales | 3.6 | 2 | 2.5 | 1.7 | 0.9 | 0.7 |
| Proteobacteria; Rhizobiales | 1.3 | 1.4 | 1.9 | 1.1 | 0.9 | 1.1 |
| Proteobacteria; Myxococcales | 1.2 | 1.1 | 1.6 | 1.1 | 1 | 1.3 |
| Chloroflexi; Thermomicrobiales | 3.7 | 4.3 | 5 | 5.5 | 7 | 9 |
| Verrucomicrobia; Chthoniobacterales | 2.4 | 1.2 | 1.8 | 1.1 | 0.9 | 1.9 |
| Gemmatimonadetes; Gemmatimonadales | 4.3 | 2.4 | 2.5 | 2.5 | 1.3 | 1.5 |
| Firmicutes; Bacillales | 0.7 | 0.6 | 1.6 | 5.3 | 3.4 | 4.7 |
| Patecibacteria; Saccharimonadales | 1.5 | 2.1 | 1.8 | 1.7 | 2.4 | 1.8 |
| Planctomyces; Pirellulales | 1.7 | 1 | 1.5 | 0.9 | 0.6 | 0.4 |

% Read Abundance

— 7.5

— 5.0

— 2.5

**Figure 7: Heatmap of the 16 most abundant bacterial orders in the analyzed soil samples grouped by N levels and litter input treatments (n=4, except for N2 Root: n=3)**