# Peer review of "Maize root and shoot litter quality controls short-term CO2 and N2O emissions and bacterial community structure of arable soil"

_Biogeosciences, 2019_

## Referee Comment (RC1) · Anonymous Referee #1 · 29 Sep 2019

The manuscript submitted by Rummel and coworkers for publication in Biogeoscienes describes the role of litter quality for N2O as well as CO2 emissions as well as bacterial community structure. The authors used litter material from maize roots and shoots which were grown under different fertilization levels, applied the materials in a pot experiment to soil which was obtained from an agricultural field and measured for a period of 22 days gas fluxes as well as chemical parameters. At the end of the incubation period also bacterial community structure was analysed. As expected depending on the C:N ratio of the litter material and the availability of easily degradable materials gas emissions and N pools in soil changed, which was also reflected by shifts in bacterial community structure. The study is nicely performed and the data presented of interest,

although not totally new. The paper is nicely written and the figures are clear. Like always in such experiments, there is the issue of water content, which was fixed to 50 % max WHK, however other water contents would for sure change the results (mainly fluctuation water levels like observed in the field) and also the use of other soil types may induce different response patter. I think here the discussion must be adapted accordingly to make sure that this is showcase but not a general response. Furthermore there are several issues that need to be considered during revision 1. The description of the sequencing data is very poor. Neither basic data on reads quality rarefraction subsampling etc is given, nor analysis of core microbiomes (together with responders) were made. I guess this is somehow a missed change and the paper would much benefit from a better integration of the molecular data. Further the sequencing data needs to be submitted to a public database. Finally it is general accepted that all DNA extraction kits contain contaminating DNA. Thus a water extraction control would be essential to remove contaminating OTUs from the data. 2. I miss data on bacterial abundance microbial biomass C and N etc. This information is required and the one hand as soil microbes are an important storage device for N. On the other hand all molecular data is relative, thus to translate the data to absolute numbers biomass values are needed. 3. I am quite confused that only three replicates were used for molecular analysis, despite 4 replicates were used for each treatment. Further I wonder why only shoots from N2 were used and not shoots from N1 treatment. 4. The provided hypothesis is very generic and I guess it must be specified as it is quite obvious that the degree of label materials influences process rates in soil.
* * *

---

## Referee Comment (RC2) · Anonymous Referee #2 · 18 Oct 2019

It is a thoroughly conducted study and a well written manuscript, that warrants publication. I have several only minor questions and requests for some clarifications: 1) My biggest confusion when reading the manuscript was the regression analyses between the emissions and the amounts of added litter. Were not the same amounts of litter with the same properties added to each treatment? If that is correct, with only two treatments, how is it possible to do a regression? If that is not correct, better explanations are needed in the Methods. 2) There is a need to describe the reasoning for some of the experimental choices and decisions that the authors made. a. What was the purpose of growing plants at two different N rates? I presumed that since you had plants grown at two different N levels you would use their litter separately. If the

point was that the plants grown at two different rates will generate different N levels in the soil, would it not be just easier to add N to the soil prior to the incubation? b. Why the samples were not just incubated in the dark as, commonly done? 3) Some improvement in organization might be warranted. Section 2.2 - I would start the section with a general description of the experiment (what is currently located on ll. 119-120); then add the specific details about shoot and root plant preparations later. As is, it is confusing. 4) Minor suggestions: a. L.273-274 – this information will be more visible when reported in a table, instead of being buried in the text. b. In some places you talk about statistical significance and provide p-values, in others you say how things are different but without mentioning the statistical significance. I suggest being consistent and either only talk about statistically significant differences or specify what is being regarded as numeric and what as statistically significant difference. c. L. 351-354 and l. 368-370 – I don' believe that just the correlation results can warrant the conclusions that are stated in these two cases.

---

## Author Comment (AC1) · 14 Nov 2019

We are grateful for the positive evaluation of our work and the constructive suggestions to improve the quality of our manuscript. In the following, we reply as comprehensively as possible to each of the comments:

*The manuscript submitted by Rummel and coworkers for publication in Biogeoscienes describes the role of litter quality for N2O as well as CO2 emissions as well as bacterial community structure. The authors used litter material from maize roots and shoots which were grown under different fertilization levels, applied the materials in a pot experiment to soil which was obtained from an agricultural field and measured for a period of 22 days gas fluxes as well as chemical parameters. At the end of the incubation period also bacterial community structure was analysed. As expected depending on the C:N ratio of the litter material and the availability of easily degradable materials gas emissions and N pools in soil changed, which was also reflected by shifts in bacterial community structure. The study is nicely performed and the data presented of interest, although not totally new. The paper is nicely written and the figures are clear. Like always in such experiments, there is the issue of water content, which was fixed to 50% max WHK, however other water contents would for sure change the results (mainly fluctuation water levels like observed in the field) and also the use of other soil types may induce different response patter. I think here the discussion must be adapted accordingly to make sure that this is showcase but not a general response.*

We agree that soil moisture is an important control of $N_2O$ emissions and changes in water content would affect results. Certainly, fluctuations of water content would induce different response patterns. We will address this topic in a respective paragraph in the discussion:

In addition to soil mineral N concentration and plant litter, soil type and soil moisture may have influenced our results (e.g. Aulakh et al., 1991). Increasing soil moisture leads to increasing $N_2O$ emissions, but relative contribution of nitrification and denitrification to $N_2O$ formation may change with increasing soil moisture (Bateman and Baggs, 2005; Baral et al., 2016; Li et al., 2016). Therefore, future experiments with different soil moisture contents should include methods to differentiate between $N_2O$ formation pathways.

*Furthermore there are several issues that need to be considered during revision 1. The description of the sequencing data is very poor. Neither basic data on reads quality rarefraction subsampling etc is given*

In addition to the information you find in the manuscript, we included raw reads, reads after filtering, subsample size, observed ASVs, and diversity indices in Table S4 in the Supplementary that was published alongside the manuscript. Rarefaction curves of the observed

amplicon sequence variants (ASVs) of the soil inhabiting bacterial communities are displayed in Supplementary Fig. S1.

*nor analysis of core microbiomes (together with responders) were made. I guess this is somehow a missed change and the paper would much benefit from a better integration of the molecular data.*

As suggested, we analyzed the core microbiomes and respective responders and will include the following paragraphs in the manuscript. Due to size limitations core microbiome tables and Venn diagrams will be included in the Supplementary.

**2.5.2 Sequence processing**

The core microbiomes and respective responders have been analyzed on genus level, grouped by either the applied litter treatment or N fertilizer levels using the R package *ampvis2* v2.4.7.

**3.5 Bacterial community structure**

The most abundant genera attributed to the core microbiomes were *Pseudomonas*, *Altererythrobacter*, *Gaiella*, *Nocardioides*, *Agromyces*, *Bacillus*, and *Lysobacter*. Overall, 80 genera were attributed to the core microbiome, when grouped by N levels, while 21 genera and 6 genera were identified as responders to N1 and N2, respectively. In detail, the responders to the applied N treatment were, among others, the genera *Chthonibacter*, *Luteimonas*, *Sphingobium*, *Novosphingobium*, *Adhaeribacter*, *Nitrospira*, *Gemmata* and *Devosia* for N1 and *Conexibacter* for N2 samples. When grouped by litter treatment, the core microbiome comprised 77 genera accounting for 73% of the relative abundance, while 9, 3 and 10 genera were identified as responders to the applied litter treatment Control, Root and Root+Shoot, respectively. *Nonomuraea*, *Fluviicola* and *Nitrospira* responded to the Root+Shoot treatment, while the genera *Lapillicoccus* and *Adhaeribacter* responded to the Root treatment. The genera *Litorilinea*, *Gemmata*, *Novosphingobium* and *Opitutus* were identified as responders to the Control treatment. For N levels and litter treatments respectively, 833 and 838 genera were attributed to non-core microbiomes, accounting for 20% and 19.5% of relative abundance.

**4.3 Bacterial community structures as affected by maize litter and soil N level**

The most abundant phyla in our soil samples, *Actinobacteria*, *Proteobacteria* and *Chloroflexi,* were also affiliated to the core microbiomes. […]

*Actinobacteria*, *Chloroflexi*, and *Firmicutes* were more abundant in N2 samples, whereas *Bacteroidetes*, and *Nitrospirae* were more abundant in N1 samples which may indicate that the latter are more competitive under conditions of very low mineral nitrogen availability in soil. This

was further validated as *Nitrospira* (*Nitrospirae*), known to oxidize nitrite (Koch et al., 2015), was identified as a responder for N1 and -RS. […]

Species belonging to the genus *Agromyces* (*Actinobacteria*), which was affiliated to the core microbiomes, are also known to reduce nitrate (Zgurskaya et al., 2008). In addition, species capable of denitrification under anaerobic, $O_2$-limited and aerobic conditions can be found in the genera *Bacillus* and *Micromonospora*, as well as *Pseudomonas* and *Rhodococcus* (Verbaendert et al., 2011) that were affiliated to the core microbiome but were more abundant in N2 samples. The genus *Opitutus* was identified as responder to -Cn and comprises the bacterium *Optitutus terrae* that was only found in anoxic habitats in soils (Chin et al., 2001). […]

*Further the sequencing data needs to be submitted to a public database.*

The information on sequence data availability can be found under "Data availability" in line 449: "The 16S rRNA gene sequences were deposited in the National Centre for Biotechnology Information (NCBI) Sequence Read Archive (SRA) under bioproject number PRJNA557843."

*Finally it is general accepted that all DNA extraction kits contain contaminating DNA. Thus a water extraction control would be essential to remove contaminating OTUs from the data.*

As described in line 164ff, we did not use a commercial DNA extraction kit. Instead the DNA extraction protocol by Griffiths et al. (2000) was used. All solutions were sterilized by either autoclaving or sterile filtration. In addition, controls for contamination were carried out during the extractions and in subsequent PCRs.

*2. I miss data on bacterial abundance microbial biomass C and N etc. This information is required and the one hand as soil microbes are an important storage device for N. On the other hand all molecular data is relative, thus to translate the data to absolute numbers biomass values are needed.*

We agree that it is highly likely that microbial biomass varies depending on litter input. Strong differences in $CO_2$ emissions between treatments indicate differences in microbial activity which could be reasoned by variations in microbial biomass and taxonomy. Nonetheless, the here presented study shows how microbial community composition responds to different litter inputs and whether the relative differences in microbial community structure can be related to $CO_2$ and $N_2O$ emissions. However, in the here presented work, changes of microbial biomass were not in the scope of the research objective and therefore this data was not collected.

*3. I am quite confused that only three replicates were used for molecular analysis, despite 4 replicates were used for each treatment.*

We used all four replicates of each treatment for molecular analysis. However, for one replicate of N2-Rt, DNA concentration was too low and the 16S rRNA gene PCR was not successful, thus only the remaining three replicates of this treatment were evaluated. In addition to figure and table captions, we will include this information in Material & Methods section 2.5.2.

*Further I wonder why only shoots from N2 were used and not shoots from N1 treatment.*

To be able to compare the litter treatments over soil conditions, we had to use the same litter types for both soil N levels. We will specify our choices in Materials & Methods section 2.2:

The incubation experiment consisted of a two-factorial setup comprising two N levels (N1 and N2) and three litter levels (Control = Cn, Root = Rt, Root+Shoot = RS) (see Table 1 and Figure 1 for details). To allow comparison of litter treatments over soil conditions, the same litter types for both soil N levels were used. As N2 plants had produced greater and healthier biomass during pre-experimental growth phase, only N2 shoots were used for both soils. Roots from N1 and N2 plants were mixed to ensure sufficient amounts for all replicates. […]

*4. The provided hypothesis is very generic and I guess it must be specified as it is quite obvious that the degree of label materials influences process rates in soil.*

We will specify our hypotheses as following: We hypothesize that differences in $N_2O$ emissions between treatments can be related to degradability of maize litter with easier degradable shoot litter leading to higher $N_2O$ formation. We further expect that differences in litter chemical quality are reflected in the structural composition of the soil microbial community with higher availability of N and C leading to a more specialized community.

---

## Author Comment (AC2) · 14 Nov 2019

We thank you for thoroughly reading our manuscript and the detailed and constructive comments to improve the quality of our paper. In the following, we will reply to each remark in detail:

*It is a thoroughly conducted study and a well written manuscript, that warrants publication. I have several only minor questions and requests for some clarifications: 1) My biggest confusion when reading the manuscript was the regression analyses between the emissions and the amounts of added litter. Were not the same amounts of litter with the same properties added to each treatment? If that is correct, with only two treatments, how is it possible to do a regression? If that is not correct, better explanations are needed in the Methods.*

We used data of all 24 pots for the regression analyses. The amounts of litter added differed between the three litter treatments as described in Table 1. We agree that having only three litter levels does not allow to draw general conclusions. However, for the soil and litter used in our study, the regressions summarize the relationship between litter quality, mineralization, and $N_2O$ and $CO_2$ emissions.

We will include the missing information in L. 221-223:

For cumulative $CO_2$ emissions, regression models included the factors total C input, water-extractable C input, hemicellulose fraction, cellulose fraction, and lignin fraction from all litter treatments (-Cn, -Rt, -RS, n=24).

*2) There is a need to describe the reasoning for some of the experimental choices and decisions that the authors made. a. What was the purpose of growing plants at two different N rates? I presumed that since you had plants grown at two different N levels you would use their litter separately. If the point was that the plants grown at two different rates will generate different N levels in the soil, would it not be just easier to add N to the soil prior to the incubation?*

The main purpose of growing plants at two N rates was to obtain soils with different background mineral N levels for the incubation experiment. We did not add any fresh mineral N immediately before onset of the incubation because we wanted to simulate conditions comparable to agricultural practice in Europe where in most countries farmers are not allowed to add mineral N with crop residues/catch crops. In addition, soil microorganisms adapt to different N availability during plant growth phase.

We will specify this information in the introduction (L. 81 ff):

Maize plants were grown in a greenhouse to produce root and shoot litter. As farmers in most European countries are not allowed to add mineral N with incorporation of crop residues or catch crops, we applied two N fertilizer regimes (low vs. high) to realize differences in soil $N_{min}$

concentration at harvest. We then set up a laboratory incubation experiment with fresh maize root or root and shoot litter under fully controlled conditions and determined hourly $CO_2$ and $N_2O$ fluxes for 22 days.

We decided to use a two-factorial design for the incubation experiment. Thus, we used the same litter types for both soil N levels to be able to compare the litter treatments over soil conditions. We will clarify this in Material and Methods section 2.2 (see improved section 2.2 below).

*b. Why the samples were not just incubated in the dark as, commonly done?*

We agree that the information given in L. 132 ff is misleading and will be corrected: The samples were covered with PVC lids, to minimize evaporation from the soil and to incubate samples in the dark.

*3) Some improvement in organization might be warranted. Section 2.2 - I would start the section with a general description of the experiment (what is currently located on ll. 119-120); then add the specific details about shoot and root plant preparations later. As is, it is confusing.*

We improved this section according to your suggestions starting with a general description of the experimental design and explanation of the experimental choices. Then, we describe preparations of treatments and setting up of the experiment:

L. 115-135

**2.2 Incubation experiment**

The incubation experiment consisted of a two-factorial setup comprising two N levels (N1 and N2) and three litter levels (Control = Cn, Root = Rt, Root+Shoot = RS) (see Table 1 and Figure 1 for details). To allow comparison of litter treatments over soil conditions, the same litter types for both soil N levels were used. As N2 plants had produced greater and healthier biomass during pre-experimental growth phase, only N2 shoots were used for both soils. Roots from N1 and N2 plants were mixed to ensure sufficient amounts for all replicates. Control soils (N1-Cn and N2-Cn) did not receive plant biomass, yet they contained C input from rhizodeposition of the previous maize growth. C remaining from rhizodeposition, root hairs and small root fragments was calculated as the difference in soil C concentration before and after maize growth. For the root treatment, 100 g fresh root biomass was added per kg dry soil (N1-Rt and N2-Rt), and in the root and shoot treatment, 100 g fresh root and 100 g fresh shoot biomass was added per kg dry soil (N1-RS, N2-RS). Each treatment was replicated four times.

Within each N level, soil was homogenized to ensure similar starting conditions. Subsamples of both soils were taken for analysis of mineral N, water extractable $C_{org}$ concentration, and total

soil C. Soil mineral N concentrations were 0.93 and 1.97 mg N kg$^{-1}$ for N1 and N2, respectively. Plant litter was cut to a size of 2 cm and homogeneously mixed with the soil, simulating residue incorporation and tillage. PVC pots with a diameter of 20 cm and a total volume of 6.8 L were filled with fresh soil equivalent to 3.5 kg dry weight previously mixed with plant litter. Soil was compacted in a stepwise mode by filling a 2 cm-layer of soil in pots and compacting it with a plunger. To ensure continuity between soil layers, the surface of the compacted layer was gently scratched before adding the next soil layer. Due to high litter input, target bulk density was 1.1 g cm$^{-3}$. Actual bulk density was determined by measuring headspace height, and these values were used for calculations.

To adjust soil moisture of all pots to 70% WHC, equivalent to 49% WFPS, water was dripped on the soil surface through hollow needles (outer diameter 0.9 mm). Pots were covered with PVC lids to minimize evaporation from the soil surface and to incubate samples in the dark. The incubation experiment was carried out under controlled temperature (16 h day at 25°C, 8 h night at 19°C) for 22 days. Volumetric water content (VWC) sensors (EC-5, Decagon Devices, Pullman, USA) were used to monitor soil water content.

*4) Minor suggestions: a. L.273-274 – this information will be more visible when reported in a table, instead of being buried in the text.*

Data on soil $NO_3^-$ and $NH_4^+$ concentrations are shown in Figures 4 a and b. We will add a table showing mineralization during the incubation period.

*b. In some places you talk about statistical significance and provide p-values, in others you say how things are different but without mentioning the statistical significance. I suggest being consistent and either only talk about statistically significant differences or specify what is being regarded as numeric and what as statistically significant difference.*

We will add p-values for differences between cumulative $CO_2$ and $N_2O$ emission in the text (L. 255 ff.). Currently, these values are depicted in Table 3. In all other cases, p-values are given in the text and in the respective tables. We did not conduct statistics on hourly $N_2O$ and $CO_2$ fluxes or soil $NO_3^-$, $NH_4^+$, and WEOC concentrations. Thus, we do not provide p-values for these.

L. 254-259

To account for different C inputs in treatments, cumulative $CO_2$ and $N_2O$ emissions were standardized against the C input per treatment (Table 1). Still, cumulative $CO_2$ emissions were almost twice as high in -Rt and about four times higher in -RS compared to -Cn (p<0.05), indicating that differences between litter treatments cannot simply be explained by differences in

C input. Addition of maize root and shoot litter increased cumulative $N_2O$ emissions by roughly 100-times compared to control treatments (p<0.05). In contrast, root litter increased cumulative $N_2O$ emissions only by a factor of 5.4 (N1-Rt) and 7 (N2-Rt) compared to the respective controls (p<0.05).

*c. L. 351-354 and l. 368-370 – I don' believe that just the correlation results can warrant the conclusions that are stated in these two cases.*

We improved these paragraphs as following:

L. 351-354

Denitrification in soil is largely controlled by the supply of readily decomposable organic matter (Azam et al., 2002; Burford and Bremner, 1975; Loecke and Robertson, 2009), leading to significant correlations between both hourly and cumulative $N_2O$ and $CO_2$ emissions (Azam et al., 2002; Fiedler et al., 2017; Frimpong and Baggs, 2010; Huang et al., 2004; Millar and Baggs, 2004, 2005). Hourly $CO_2$ fluxes increased directly with onset of incubation and started to decline after day 10, thus mostly C compounds with a short turnover time, i.e. sugars, proteins, starch, and hemicellulose were decomposed and contributed to $CO_2$ fluxes. Availability of easily degradable C compounds stimulates microbial respiration, limiting $O_2$ at the microsite level and thus increasing $N_2O$ emissions from denitrification (Azam et al., 2002; Chen et al., 2013; Miller et al., 2008). Accordingly, hourly $N_2O$ fluxes increased after a lag phase of two days. The strong positive correlation ($R^2$=0.9362, $p \leq 7.632\ e^{-15}$) between cumulative $CO_2$ and $N_2O$ emissions (Table 6) further supports our hypothesis that litter quality, in particular degradability of C compounds, affects $N_2O$ fluxes from denitrification by creating plant litter associated microsites with low $O_2$ concentrations.

L. 368-371

High correlation of cumulative $N_2O$ emissions and mineralized N during the incubation period ($R^2$=0.5791, $p<9.551\ e^{-06}$) indicates that, in addition to denitrification, heterotrophic nitrification may have contributed to $N_2O$ production in our study. However, to further differentiate between processes contributing to $N_2O$ production, stable isotope methods need to be used (Baggs, 2008; Butterbach-Bahl et al., 370 2013; Van Groenigen et al., 2015; Wrage-Mönnig et al., 2018).

---

## Author Response (AR1)

**Authors response**

We thank the Editor for the opportunity to improve the quality of our manuscript and his valuable feedback. We considered all comments and remarks by the editor and the referees and improved our manuscript accordingly. In addition, we clarified some phrasing, corrected typos, and revised punctuation.

**Response to Editor's comments:**

*When revising your manuscript I think you should at least discuss how information on microbial biomass would add to an interpretation of your findings. It is not enough to simply state that they were not in the scope of the research objective.*

We added a paragraph explaining how analysis of microbial biomass could give further insights in lines 402-409:

Under N limiting conditions, a higher portion of N is recovered in soil microbial biomass in relation to litter N input (Bending and Turner, 1999, Troung and Marschner, 2018). When N is abundant relative to C availability, excess N is released by soil microorganisms and can be lost as $N_2O$. In -Rt, where N availability was low, N was immobilized by soil microorganisms and $N_2O$ emission were low. When more easily degradable N was added with maize shoots, N released from decomposition of maize shoots presumably fostered decomposition of maize roots (Robertson and Groffman, 2015) and denitrification of excess N leading to strongly increased $CO_2$ and $N_2O$ emissions in -RS. To estimate the contribution of plant litter N to mineralization, immobilization, and denitrification, $^{15}N$ labeled litter together with analysis of microbial biomass N and $^{15}N_2O$ emissions could be used (e.g. Frimpong and Baggs, 2010; Ladd et al., 1981).

*In addition to the comments by the reviewers I would like to strongly encourage you to use SI-Units throughout your manuscript (including the Figures), eg. a flux should be expressed per g s instead of per kg h.*

We changed figures 2 and 3 a+b to show fluxes per g s and updated the description accordingly.

**Response to Referee Comment #1, uploaded November 14th, 2019:**

Updates (e.g. line numbers in **bold**).

We are grateful for the positive evaluation of our work and the constructive suggestions to improve the quality of our manuscript. In the following, we reply as comprehensively as possible to each of the comments:

*The manuscript submitted by Rummel and coworkers for publication in Biogeoscienes describes the role of litter quality for N2O as well as CO2 emissions as well as bacterial community structure. The*

*authors used litter material from maize roots and shoots which were grown under different fertilization levels, applied the materials in a pot experiment to soil which was obtained from an agricultural field and measured for a period of 22 days gas fluxes as well as chemical parameters. At the end of the incubation period also bacterial community structure was analysed. As expected depending on the C:N ratio of the litter material and the availability of easily degradable materials gas emissions and N pools in soil changed, which was also reflected by shifts in bacterial community structure. The study is nicely performed and the data presented of interest, although not totally new. The paper is nicely written and the figures are clear. Like always in such experiments, there is the issue of water content, which was fixed to 50% max WHK, however other water contents would for sure change the results (mainly fluctuation water levels like observed in the field) and also the use of other soil types may induce different response patter. I think here the discussion must be adapted accordingly to make sure that this is showcase but not a general response.*

We agree that soil moisture is an important control of $N_2O$ emissions and changes in water content would affect results. Certainly, fluctuations of water content would induce different response patterns. We **addressed** this topic in a respective paragraph in the discussion **(L. 426-430)**:

In addition to soil mineral N concentration and plant litter, soil type and soil moisture may have influenced our results (e.g. Aulakh et al., 1991). Increasing soil moisture leads to increasing $N_2O$ emissions, but relative contribution of nitrification and denitrification to $N_2O$ formation may change with increasing soil moisture (Bateman and Baggs, 2005; Baral et al., 2016; Li et al., 2016). Therefore, future experiments with different soil moisture contents should include methods to differentiate between $N_2O$ formation pathways.

*Furthermore there are several issues that need to be considered during revision 1. The description of the sequencing data is very poor. Neither basic data on reads quality rarefraction subsampling etc is given*

In addition to the information you find in the manuscript, we included raw reads, reads after filtering, subsample size, observed ASVs, and diversity indices in Table S4 in the Supplementary that was published alongside the manuscript. Rarefaction curves of the observed amplicon sequence variants (ASVs) of the soil inhabiting bacterial communities are displayed in Supplementary Fig. S1.

*nor analysis of core microbiomes (together with responders) were made. I guess this is somehow a missed change and the paper would much benefit from a better integration of the molecular data.*

As suggested, we analyzed the core microbiomes and respective responders and **included** the following paragraphs in the manuscript. Core microbiome tables (**Table S6-S8**) and Venn diagrams (Fig. S5) **are** included in the Supplementary.

2.5.2 Sequence processing

In addition, core microbiomes and respective responders have been analyzed on genus level, grouped by either the applied litter treatment or N fertilizer levels using *ampvis2* v2.4.7 (**L. 211-212**).

3.5 Bacterial community structure

At the genus level, *Pseudomonas*, *Altererythrobacter*, *Gaiella*, *Nocardioides*, *Agromyces*, *Bacillus,* and *Lysobacter* were most abundant accounting for up to 5.7 % of all ASVs. Accordingly, these were also most abundant genera attributed to the core micorbiome (Tables S6 and S8). Overall, 80 genera represented the core microbiome, when grouped by N levels, while 21 genera and 6 genera were identified as responders to N1 and N2, respectively (Fig. S5). In detail, the classified responders to the applied N treatments were the genera *Chthonibacter*, *Luteimonas*, *Sphingobium*, *Novosphingobium*, *Adhaeribacter*, *Nitrospira*, *Gemmata*, and *Devosia* for N1 and *Conexibacter* for N2 samples (Table S. 8). The genera *Bacillus*, *Gaiella*, *Altererythrobacter*, *Blastococcus*, and *Pseudomonas* showed highest abundance in N2 samples, while *Lysobacter*, and *Sphingomonas* were more abundant in N1 samples (Fig. S3). When grouped by litter treatment, the core microbiome comprised 77 genera accounting for 73 % of the relative abundance, while 9, 3 and 10 genera were identified as responders to the applied litter treatments Control, Root and Root+Shoot, respectively (Fig. S5). *Nonomuraea*, *Fluviicola*, and *Nitrospira* responded to the Root+Shoot treatment, while the genera *Lapillicoccus* and *Adhaeribacter* responded to the Root treatment (Table S7). The genera *Litorilinea*, *Gemmata*, *Novosphingobium*, and *Opitutus* were identified as responders to the Control treatment. For N levels and litter treatments respectively, 833 and 838 genera were identified as non-core microbiomes, accounting for 20 % and 19.5 % of relative abundance (Fig. S5) (**L. 316-330**).

4.3 Bacterial community structures as affected by maize litter and soil N level

The most abundant phyla in our soil samples were the *Actinobacteria*, *Proteobacteria*, and *Chloroflexi*. Among these phyla, the genera *Pseudomonas* (*Proterobacteria*) and *Gaiella* (*Actinobacteria)* were also affiliated to the core microbiomes (**L. 444-445**). […]

In our treatments, *Actinobacteria*, *Chloroflexi*, and *Firmicutes* were more abundant in N2 samples, whereas *Bacteroidetes*, and *Nitrospirae* were more abundant in N1 samples which may indicate that the latter are more competitive under conditions of very low mineral nitrogen availability in soil. This was further validated as *Nitrospira* (*Nitrospirae*), known to oxidize nitrite (Koch et al., 2015), was identified as a responder for N1 and -RS (**L. 466-469**). […]

Species belonging to the genus *Agromyces* (*Actinobacteria*), which was affiliated to the core microbiomes, are also known to reduce nitrate (Zgurskaya et al., 2008). In addition, species capable of denitrification under anaerobic, $O_2$-limited and aerobic conditions can be found in the genera *Bacillus* and *Micromonospora*, as well as *Pseudomonas* and *Rhodococcus* (Verbaendert et al., 2011) that were affiliated to the core microbiome but were more abundant in N2 samples. The genus *Opitutus* was identified as responder to -Cn and comprises the bacterium *Optitutus terrae* that was only found in anoxic habitats in soils (Chin et al., 2001) (**L. 471-476**). […]

*Further the sequencing data needs to be submitted to a public database.*

The information on sequence data availability can be found under "Data availability" in line **490**: "The 16S rRNA gene sequences were deposited in the National Centre for Biotechnology Information (NCBI) Sequence Read Archive (SRA) under bioproject number PRJNA557843."

*Finally it is general accepted that all DNA extraction kits contain contaminating DNA. Thus a water extraction control would be essential to remove contaminating OTUs from the data.*

As described in line **169**ff, we did not use a commercial DNA extraction kit. Instead the DNA extraction protocol by Griffiths et al. (2000) was used. All solutions were sterilized by either autoclaving or sterile filtration. In addition, controls for contamination were carried out during the extractions and in subsequent PCRs.

*2. I miss data on bacterial abundance microbial biomass C and N etc. This information is required and the one hand as soil microbes are an important storage device for N. On the other hand all molecular data is relative, thus to translate the data to absolute numbers biomass values are needed.*

We agree that it is highly likely that microbial biomass varies depending on litter input. Strong differences in $CO_2$ emissions between treatments indicate differences in microbial activity which could be reasoned by variations in microbial biomass and taxonomy. Nonetheless, the here presented study shows how microbial community composition responds to different litter inputs and whether the relative differences in microbial community structure can be related to $CO_2$ and $N_2O$ emissions. However, in the here presented work, changes of microbial biomass were not in the scope of the research objective and therefore this data was not collected. **Nevertheless, we included a paragraph describing how microbial biomass could contribute to interpretation in lines 402-409.**

Under N limiting conditions, a higher portion of N is recovered in soil microbial biomass in relation to litter N input (Bending and Turner, 1999, Troung and Marschner, 2018). When N is abundant relative to C availability, excess N is released by soil microorganisms and can be lost as $N_2O$. In -Rt, where N availability was low, N was immobilized by soil microorganisms and $N_2O$ emission were low. When more easily degradable N was added with maize shoots, N released from decomposition of maize shoots presumably fostered decomposition of maize roots (Robertson and Groffman, 2015) and denitrification of excess N in -RS led to strongly increased $CO_2$ and $N_2O$ emissions. To estimate the contribution of plant litter N to mineralization, immobilization, and denitrification, $^{15}N$ labeled litter together with analysis of microbial biomass N and $^{15}N_2O$ emissions could be used (e.g. Frimpong and Baggs, 2010; Ladd et al., 1981).

*3. I am quite confused that only three replicates were used for molecular analysis, despite 4 replicates were used for each treatment.*

We used all four replicates of each treatment for molecular analysis. However, for one replicate of N2-Rt, DNA concentration was too low and the 16S rRNA gene PCR was not successful, thus only the remaining three replicates of this treatment were evaluated. In addition to figure and table captions, we included this information in Material & Methods section 2.5.2: **For one replicate of N2-Rt, DNA concentration was very low and the 16S rRNA gene could not be amplified. Thus, we only evaluated the remaining three replicates of this treatment (L. 213-214).**

*Further I wonder why only shoots from N2 were used and not shoots from N1 treatment.*

To be able to compare the litter treatments over soil conditions, we had to use the same litter types for both soil N levels. We **specified** our choices in Materials & Methods section 2.2 (**L. 118-122**):

The incubation experiment consisted of a two-factorial setup comprising two N levels (N1 and N2) and three litter levels (Control = Cn, Root = Rt, Root+Shoot = RS) (see Table 1 and Figure 1 for details). To allow comparison of litter treatments over soil conditions, the same litter types for both soil N levels were used. As N2 plants had produced greater and healthier biomass during pre-experimental growth phase, only N2 shoots were used for both soils. Roots from N1 and N2 plants were mixed to ensure sufficient amounts for all replicates. […]

*4. The provided hypothesis is very generic and I guess it must be specified as it is quite obvious that the degree of label materials influences process rates in soil.*

We **specified** our hypotheses as following: We hypothesize that differences in $N_2O$ emissions between treatments can be related to degradability of maize litter with easier degradable shoot litter leading to higher $N_2O$ formation. We further expect that differences in litter chemical quality are reflected in the structural composition of the soil microbial community with higher availability of N and C leading to a more specialized community (L. **78-82**).

**Response to referee comment #2, uploaded November 14th, 2019**
Updates (e.g. line numbers in **bold**).

We thank you for thoroughly reading our manuscript and the detailed and constructive comments to improve the quality of our paper. In the following, we will reply to each remark in detail:

*It is a thoroughly conducted study and a well written manuscript, that warrants publication. I have several only minor questions and requests for some clarifications: 1) My biggest confusion when reading the manuscript was the regression analyses between the emissions and the amounts of added litter. Were not the same amounts of litter with the same properties added to each treatment? If that is correct, with only two treatments, how is it possible to do a regression? If that is not correct, better explanations are needed in the Methods.*

We used data of all 24 pots for the regression analyses. The amounts of litter added differed between the three litter treatments as described in Table 1. We agree that having only three litter levels does not allow to draw general conclusions. However, for the soil and litter used in our study, the regressions summarize the relationship between litter quality, mineralization, and $N_2O$ and $CO_2$ emissions.

We **included** the missing information in L. **228-230**:
For cumulative $CO_2$ emissions, regression models included the factors total C input, water-extractable C input, hemicellulose fraction, cellulose fraction, and lignin fraction from all litter treatments (-Cn, -Rt, -RS, n=24).

*2) There is a need to describe the reasoning for some of the experimental choices and decisions that the authors made. a. What was the purpose of growing plants at two different N rates? I presumed that since you had plants grown at two different N levels you would use their litter separately. If the point was that the plants grown at two different rates will generate different N levels in the soil, would it not be just easier to add N to the soil prior to the incubation?*

The main purpose of growing plants at two N rates was to obtain soils with different background mineral N levels for the incubation experiment. We did not add any fresh mineral N immediately before onset of the incubation because we wanted to simulate conditions comparable to agricultural practice in Europe where in most countries farmers are not allowed to add mineral N with crop residues/catch crops. In addition, soil microorganisms adapt to different N availability during plant growth phase.

We **specified** this information in the introduction (L. **84**ff):
Maize plants were grown in a greenhouse to produce root and shoot litter. As **in many European countries law prohibits addition of** mineral N with incorporation of crop residues or catch crops, we applied two N fertilizer regimes (low vs. high) to realize differences in soil $N_{min}$ concentration at harvest. We then set up a laboratory incubation experiment with fresh maize root or root and shoot litter under fully controlled conditions and determined hourly $CO_2$ and $N_2O$ fluxes for 22 days.

We decided to use a two-factorial design for the incubation experiment. Thus, we used the same litter types for both soil N levels to be able to compare the litter treatments over soil conditions. We will clarify this in Material and Methods section 2.2 (see improved section 2.2 below).

*b. Why the samples were not just incubated in the dark as, commonly done?*

We agree that the information given in L. **137**ff **was** misleading and **corrected** it accordingly:

**Pots** were covered with PVC lids, to minimize evaporation from the soil and to incubate samples in the dark.

*3) Some improvement in organization might be warranted. Section 2.2 - I would start the section with a general description of the experiment (what is currently located on II. 119-120); then add the specific details about shoot and root plant preparations later. As is, it is confusing.*

We improved this section according to your suggestions starting with a general description of the experimental design and explanation of the experimental choices. Then, we describe preparations of treatments and setting up of the experiment:

L. **117-140:**

**2.2 Incubation experiment**

The incubation experiment consisted of a two-factorial setup comprising two N levels (N1 and N2) and three litter levels (Control = Cn, Root = Rt, Root+Shoot = RS) (see Table 1 and Figure 1 for details). To allow comparison of litter treatments over soil conditions, the same litter types for both soil N levels were used. As N2 plants had produced greater and healthier biomass during pre-experimental growth phase, only N2 shoots were used for both soils. Roots from N1 and N2 plants were mixed to ensure sufficient amounts for all replicates. Control soils (N1-Cn and N2-Cn) did not receive plant biomass, yet they contained C input from rhizodeposition of the previous maize growth. C remaining from rhizodeposition, root hairs and small root fragments was calculated as the difference in soil C concentration before and after maize growth. For the root treatment, 100 g fresh root biomass was added per kg dry soil (N1-Rt and N2-Rt), and in the root and shoot treatment, 100 g fresh root and 100 g fresh shoot biomass was added per kg dry soil (N1-RS, N2-RS). Each treatment was replicated four times.

Within each N level, soil was homogenized to ensure similar starting conditions. Subsamples of both soils were taken for analysis of mineral N, water extractable $C_{org}$ concentration, and total soil C. Soil mineral N concentrations were 0.93 and 1.97 mg N kg$^{-1}$ for N1 and N2, respectively. Plant litter was cut to a size of 2 cm and homogeneously mixed with the soil, simulating residue incorporation and tillage. PVC pots with a diameter of 20 cm and a total volume of 6.8 L were filled with fresh soil equivalent to 3.5 kg dry weight previously mixed with plant litter. Soil was compacted in a stepwise mode by filling a 2 cm-layer of soil in pots and compacting it with a plunger. To ensure continuity between soil layers, the surface of the compacted layer was gently scratched before adding the next soil layer. Due to high litter input, target bulk density was 1.1 g cm$^{-3}$. Actual bulk density was determined by measuring headspace height, and these values were used for calculations.

To adjust soil moisture of all pots to 70 % WHC, equivalent to 49 % WFPS, water was dripped on the soil surface through hollow needles (outer diameter 0.9 mm). Pots were covered with PVC lids to minimize evaporation from the soil surface and to incubate samples in the dark. The incubation experiment was carried out under controlled temperature (16 h day at 25 °C, 8 h night at 19 °C) for 22 days. Volumetric water content (VWC) sensors (EC-5, Decagon Devices, Pullman, USA) were used to monitor soil water content.

*4) Minor suggestions: a. L.273-274 – this information will be more visible when reported in a table, instead of being buried in the text.*

Data on soil $NO_3^-$ and $NH_4^+$ concentrations are shown in Figures 4 a and b. We **added** a table showing mineralization during the incubation period (**Table 4**).

*b. In some places you talk about statistical significance and provide p-values, in others you say how things are different but without mentioning the statistical significance. I suggest being consistent and either only talk about statistically significant differences or specify what is being regarded as numeric and what as statistically significant difference.*

We **added** p-values for differences between cumulative $CO_2$ and $N_2O$ emission in the text (L. **261**ff.). Currently, these values are depicted in Table 3. In all other cases, p-values are given in the text and in the respective tables. We did not conduct statistics on hourly $N_2O$ and $CO_2$ fluxes or soil $NO_3^-$, $NH_4^+$, and WEOC concentrations. Thus, we do not provide p-values for these.

L. **261-266**

To account for different C inputs in treatments, cumulative $CO_2$ and $N_2O$ emissions were standardized against the C input per treatment (Table 1). Still, cumulative $CO_2$ emissions were almost twice as high in -Rt and about four times higher in -RS compared to -Cn ($p<0.05$), indicating that differences between litter treatments cannot simply be explained by differences in C input. Addition of maize root and shoot litter increased cumulative $N_2O$ emissions by roughly 100-times compared to control treatments ($p<0.05$). In contrast, root litter increased cumulative $N_2O$ emissions only by a factor of 5.4 (N1-Rt) and 7 (N2-Rt) compared to the respective controls ($p<0.05$).

*c. L. 351-354 and l. 368-370 – I don' believe that just the correlation results can warrant the conclusions that are stated in these two cases.*

We improved these paragraphs as following:

L. 351-354 **now L. 366-376**

[revised manuscript text omitted]

---

## Author Response (AR2)

**Authors' response to minor revision**

*I kindly ask you to upload a final revised manuscript, for which you have carefully checked the consistency of all units used (e.g. the units for N2O fluxes are currently not consistent between Table 5, Fig. 3 and Fig. 5). I furthermore suggest you to add, both to the conclusions section and the abstract, a concluding sentence highlighting the broader implications of your study.*

We updated Table 1, 3, 4, and 5, as well as Figures 4 and 5 to present all results per g soil. For the methods section, we'll still refer to kg soil as we did the experiment with several kg soil per incubation vessel.

In addition, we included a general sentence in Abstract and Conclusions stating that "
[revised manuscript text omitted]